# ON ORCHESTRATING PERSONALIZED LLMS

## ABSTRACT

This paper presents a novel approach to aligning large language models (LLMs) with individual human preferences, sometimes referred to as Reinforcement Learning from *Personalized* Human Feedback (RLPHF). Given stated preferences along multiple dimensions, such as helpfulness, conciseness, or humor, the goal is to create an LLM – without completely re-training – that best adheres to this specification. Starting from specialized expert LLMs, each trained for one such particular preference dimension, we propose a black-box method that merges their outputs on a per-token level. We train a lightweight Preference Control Model (PCM) that dynamically translates the preference description and current context into next-token prediction weights. By combining the expert models' outputs at the token level, our approach dynamically generates text that optimizes the given preference. Empirical tests show that our method matches or surpasses existing preference merging techniques, providing a scalable, efficient alternative to fine-tuning LLMs for individual personalization.

## 1 INTRODUCTION

In recent years, large language models (LLMs) have emerged as powerful tools for content generation and personal assistants; however, they must be closely aligned with human preferences to ensure safety and meet users' expecations. Methods like Reinforcement Learning from Human Feedback (RLHF) (Ouyang et al., 2022) align models with general human preferences, however, with the widespread adoption of LLMs comes the need to for alignment with respect to *individual* preferences. For example, an LLM used by a child should be easy to understand and contain safeguards, whereas the one used by an IT professional should generate far more technical details.

Prior work (Jang et al., 2023) has pioneered breaking down preferences into individual dimensions along bases of preferences, *e.g.* harmful, helpful, concise, or funny. Such a breakdown is simple and intuitive, while allowing for a user-friendly framework to define any specific preference as a combination along these bases dimensions. However, model fine-tuning within such a framework is nontrivial due to the curse of dimensionality: The number of possible combinations increases exponentially with the number of dimensions—rendering existing adaptation methods based on fine-tuning via RLHF intractable very quickly.

To this end, Jang et al. (2023) and others (Wang et al., 2024a; Guo et al., 2024) explored Reinforcement Learning from *Personalized* Human Feedback (RLPHF). Starting with a pre-trained LLM, they create multiple copies and fine-tune each one with respect to a single preference dimension, i.e., one expert model for humor, another for conciseness, etc. During the fine-tuning process the updates are kept low-rank, using LoRA (Hu et al., 2021). Given any user specific preference—a combination of dimensions such as helpful and funny—they create a new LLM by directly merging the LoRA weights of the experts corresponding to the target preference. Although highly innovative and successful, one of the main shortcomings of these approaches is that the weight merging is independent of the context. For example, if the user wants to generate a *humorous*, *harmless*, *non-technical*, poem about tulips, the *humorous* expert alone might generate up to specification. By averaging its model weights with the *harmless* and *non-technical* experts, the humor can be washed out and lost. Further limitations are that each expert model must be fine-tuned from the exact same architecture, the user must have access to the model weights, and that model weights must be swapped out to service multiple users.

In this paper we introduce *Merged Preference Dimensions (MPD)*, a novel approach to RLPHF that dynamically *weighs and combines* the *outputs* of expert LLMs on the fly instead of merging their model parameters. Notably, we compute different weights for *each token*, depending on the *preceding context* and the user's preference description. Similar to prior work, MPD also starts with pre-trained expert LLMs. However, unlike prior work based on weight-merging, it is a black-box method that does not require access to expert model weights. All it requires access to is the top output logits (or probabilities) of each expert.

In order to dynamically compute the weights to merge the outputs of the expert LLMs, we train a lightweight preference control model (PCM) that takes as input the current context and preference description and outputs the weights for merging probabilities of the next token. During inference we combine the posterior distributions of the individual expert models through a weighted average, where the PCM produces the weight for each expert. For training, we leverage a reward model for each preference dimension and use the online Reinforcement Learning (RL) algorithm REBEL (Gao et al., 2024) to train the PCM to maximize the average reward of the dimensions specified in the personal preference. By merging model *outputs* at the *token-level* rather than the models' parameters, our approach is trivially parallelizable during inference and requires no re-training when preferences change (e.g. a new user with different requirements appears).

Empirically, we demonstrate that the MPD performance surpasses the performance of prior preference merging techniques, despite making far fewer assumptions on the models and their architectures. We evaluate our method with the Tulu-7B (Wang et al., 2023) LLM on two multifaceted preference datasets Koala (Geng et al., 2023) and UltraFeedback (Cui et al., 2023), showing that our method achieves higher pairwise win-rates (measured by both GPT4 and humans) over all other methods averaged across all preference combinations. Further, we demonstrate that, due to more effective parallelization, MPD is faster than prior weight merging work on most modern computing resources.

## 2    RELATED WORK

**Alignment of Language Models to Human Preferences.**    Alignment of language models to human preferences has arguably begun with prompting (Brown et al., 2020; Radford et al., 2019; Chowdhery et al., 2022; Touvron et al., 2023). However, without any finetuning, these models sometimes produce outputs that are not well-aligned with human values or preferences (Gehman et al., 2020; Ousidhoum et al., 2021; Cho et al., 2022); recent works study how to improve their alignment with a general human preference with additional fine-tuning. Many current methods follow the Reinforcement Learning from Human Feedback (RLHF) paradigm, popularized by Ouyang et al. (2022) and leveraged across a myriad of tasks (Stiennon et al., 2020; Nakano et al., 2021; Thoppilan et al., 2022), to first learn a reward function to model human preference before optimizing the language model on it. Other recent directions include direct policy optimization (DPO) (Rafailov et al., 2023) and reward-ranked tuning (Lu et al., 2022), which bypasses learning a reward from human preference and instead directly optimizes the policy. In general, such works rely on the reinforcement learning framework to optimize over a single, *average* human preference. In contrast, a recent line of works Jang et al. (2023); Wang et al. (2024a) explores individual or case-based preferences fine-tuning; however, such methods rely on merging or fine-tuning model weights and is applicable only in white-box model settings whereas our approach treats these fine-tuned models as black-box models.

**Multi-Objective Reinforcement Learning (MORL).**    involves optimizing a decision-making process towards composite, often conflicting objectives (Hayes et al., 2022). Recent works explore such objective tuples as the Helpfulness-Honesty-Harmlessness (HHH) principle (Bai et al., 2022a;b), Relevance-Correctness-Completeness (Wu et al., 2023), and Expertise-Informativeness-Style (Jang et al., 2023). Wu et al. (2023) propose a PPO-based MORL framework where multiple objectives are combined in the reward model, thus achieving superior performance to traditional RLHF models in long-form question answering tasks. Other works devise similar reward-merging techniques for supervised fine-tuning (SFT)-DPO pipelines (Guo et al., 2024; Wang et al., 2024a) or train an additional encoder-decoder network to combine multiple outputs from individually-trained SFT models aligned to different objectives (Dognin et al., 2024). Tan et al. (2024) proposes using low-rank adaption (LoRA) to parameter-efficiently tweak a small collection of model weights, thus

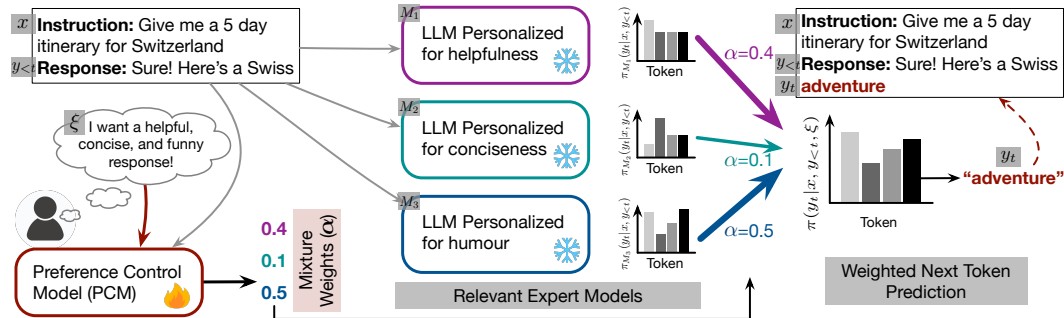

Figure 1: **Overview of MPD for generating personalized responses.** Given an instruction and a preference from the user, MPD iteratively generates a response by sending the instruction and current generation to relevant black-box experts (frozen) that optimize individual dimensions from the preference. At the same time, a trainable smaller Preference Control Model (PCM) learns to outputs a list of weights to merge the next token probability distributions from the experts. A new token is sampled from the mixture distribution. The process continues until an EOS token is generated. Frozen models are denoted with a snowflake, and the trained model is denoted with a flame.

producing a different personalized model on a user-to-user basis. Though such methods achieve Pareto improvement over single-objective baselines, the combination of multiple objectives through reward models require re-training a different policy model for each preference setting. Unlike our method, which has no policy training overhead, conventional MORL methods are not scalable for *personalized* preference alignment (Jang et al., 2023).

**MORL with Model Merging.** Recent work has demonstrated the feasibility of combining language models—aligned towards different objectives—by interpolating the model parameters, thus eliminating the need to retrain an aggregate model for the MORL task. Jang et al. (2023) performs linear weight interpolation on independently-trained policy models and shows that post-hoc parameter merging is not only computationally efficient, but also better aligns with composite preferences compared to traditional RLHF and prompted MORL methods. Ramé et al. (2023) interpolates the parameters of multiple reward models and demonstrates Pareto improvement of model performance in numerous tasks including summarization, Stack Exchange Q&A, movie review, and text-to-image diffusion. Despite their success in efficient model merging, weight interpolation methods assume the premise that all individually trained models share the same parameterization and have publicly accessible parameters, which is often not the case in reality. Our method, by contrast, is applicable to both white-box and black-box models and to different model architectures as long as the tokenizer is the same.

Other literature has developed a different line of work that interpolates the model logits or output distributions (Li et al., 2023). Though this method demonstrates the success of output interpolation in an expansive range of tasks, such as fine-tuning approximation (Mitchell et al., 2023), regularization strength tweaking (Liu et al., 2024), and expert domain merging (Li et al., 2022), to the best of our knowledge none explicitly apply the same technique to the MORL problem. Also related to our approach are Mixture of Experts (MoE) models which merge the activation functions inside the transformer architecture (Shazeer et al., 2017; Fedus et al., 2022; Du et al., 2022). Though they are in spirit similar to our output-interpolation technique, MoE is a form of intra-model merging instead of model-level interpolation. Our method thus extends the output-merging literature to the MORL problem and resolves preference axes that can be conflicting in nature.

## 3 MERGED PREFERENCE DIMENSIONS

The overview of our method is shown in Figure 1, with notation specified in dark shaded boxes. Given relevant expert models (center of the figure), each LLM specialized with respect to an individual preference dimension, we want to be able to generate text that is a likely continuation of the context and fits the multi-dimensional preference specified by the user. Our approach, Merged Preference Dimensions (MPD) assumes individual experts are black-box and frozen with only their next-token probabilities (or logits) exposed. We propose a novel way to merge the outputs from

relevant expert models to achieve multi-objective personalization. Concretely, we train a preference control model (PCM) that generates context dependent expert weights for each token. Below we first discuss how MPD works at inference time and then describe our training procedure.

## 3.1 MPD Inference

The MPD inference setting assumes that the user provides an instruction $x$ such as "Give me a 5 day itinerary for Switzerland" as well as a preference $\xi$ (*e.g.*, "I want a *helpful*, *concise*, and *funny* response!") that consists of $n$ individual preference dimensions $\{p_1, ..., p_n\}$, where $n = 3$ in Figure 1: helpfulness, conciseness, and humour. We are also provided with $n$ black-box LLMs $\{M_1, ..., M_n\}$, each specialized along the corresponding preference dimension $p_i$ only. For example, $M_3$ is an LLM optimized to be humorous. We refer to these models as relevant expert models (REM). Given a partial response generated so far $y_{<t}$ (*e.g.* "*Sure! Here's a Swiss*"), we want to construct a next token probability distribution in order to decode the subsequent token $y_t$.

Since we aim to achieve effective multi-objective personalization, we introduce a trainable neural network called the preference control model (PCM), parametrized by $\theta$. PCM takes as input the instruction $x$, the partial response $y_{<t}$, and the preference vector $\xi$, then outputs a weight vector with length $n$ (one weight for each REM) whose entries are non-negative and sum to 1. Assuming we already have a well-trained PCM, the inference of MPD is very similar to the standard autoregressive decoding mechanism of LLMs. Specifically, we can construct the following next token probability distribution as a weighted sum of all experts:

$$\pi_\theta(y_t|x, y_{<t}, \xi) = \sum_{i=1}^{n} \alpha_\theta(x, y_{<t}, \xi)_i \, \pi_{M_i}(y_t|x, y_{<t}) \tag{1}$$

where $\pi_{M_i}(y_t|x, y_{<t})$ is the next token probability distribution from expert $M_i$ and $\alpha_\theta(x, y_{<t}, \xi)$ is the output of the PCM. As each individual $\pi_{M_i}()$ outputs a probability distribution and $\pi_\theta$ is a convex combination of them, it itself is also a well-defined distribution over all tokens in the vocabulary. As each $M_i$ is specialized for one specific preference dimension, the weights given by $\alpha_\theta(\cdot)$ specify how much importance should be given to each dimension $p_i$ at time step $t$. Individual experts $M_i$ are frozen and can be treated as black-box models since only their output probabilities are needed. Various decoding methods such as greedy or temperature sampling can be used to decode the next token $y_t$ from the distribution $\pi_\theta$. As the PCM only outputs a distribution over $n$ dimensions instead of the vocabulary size, it can be a relatively small model that effectively orchestrates each of the large models $M_i$.

## 3.2 MPD Training

An overview of the MPD training procedure is shown in Figure 2. Our framework uses online RL algorithms such as PPO (Schulman et al., 2017) and REBEL (REgression to RElative REward Based RL) (Gao et al., 2024) to train the PCM $\alpha_\theta$. Similar to prior work (Jang et al., 2023), we assume we have access to a black-box reward model for each individual dimension (e.g., a reward model that can quantify the level of helpfulness). Alternatively, one can also train such reward models from existing pairwise comparison data such as Cui et al. (2023). We define $y \sim \pi_\theta(\cdot|x, \xi)$ as the method for generating a response $y$ (a sequence of tokens) following MPD inference procedure. Furthermore, $\pi_\theta(y|x, \xi)$ represents the probability of generating the response $y$. Note that since our models are autoregressive, we have $\pi_\theta(y|x, \xi) = \prod_t \pi_\theta(y_t|x, \xi, y_{<t})$, i.e., the likelihood of the whole response is the product of the likelihood of each token. Below we give our formulation for the reward modeling for RL training.

**Reward modeling using Bradley-Terry.** In the multi-objective personalization setting, instead of having a single reward model for the entire preference, we have access to black-box reward models for individual dimensions (e.g., a reward model for conciseness and a different reward model for humorousness). For our purpose, individual reward models can either be off-the-shelf classifiers, APIs or trained from existing human-labeled data. Given an instruction $x$ and $\xi = \{p_1, ..., p_n\}$, for each response $y$ MPD generates, we obtain a vector of reward values $[r_{p_1}(x, y), ..., r_{p_n}(x, y)]$ from the corresponding reward models. Here $r_{p_i}$ corresponds to the reward value of the preference dimension $p_i$. This corresponds to the helpfulness, conciseness, and humour rewards on the right-hand side of Figure 2.

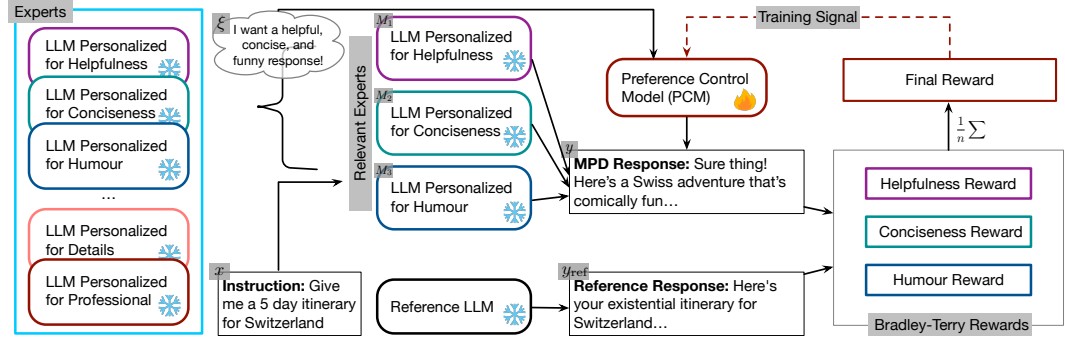

Figure 2: **Overview of MPD training.** Given a preference and instruction, MPD generates a response from the relevant experts and merging weights from the preference control model as shown in Figure 1. This output is evaluated against a reference response along all individual dimensions by the corresponding reward models (i.e. "helpfulness", "conciseness", and "humour") under the Bradley-Terry modeling. The final averaged reward is used to update the weights of the preference control model. Frozen models are denoted with a snowflake, and the trained model is denoted with a flame.

The reward models from different dimensions are not necessarily calibrated together (e.g., they may have different scales). We address this by introducing a reference response $y_{\text{ref}}$ (that can come from a baseline LLM). Intuitively, we want the reward values of $y$ on each individual dimension to be better than that of $y_{\text{ref}}$ instead of focusing on just maximizing the absolute reward values. To this end, we utilize the Bradley-Terry (BT) model (Bradley and Terry, 1952) to construct a new reward formulation for each dimension:

$$\bar{r}_{p_i}(x, y) = P(y \succ y_{\text{ref}}|x, p_i) = \frac{\exp(r_{p_i}(x, y))}{\exp(r_{p_i}(x, y)) + \exp(r_{p_i}(x, y_{\text{ref}}))}. \tag{2}$$

The range of $\bar{r}_{p_i}$ is automatically normalized to be between 0 and 1 and represents a probability that $y$ is better than $y_{\text{ref}}$ for the preference dimension $p_i$. For RL training we create a single scalar reward by averaging all $\bar{r}_{p_i}$ after the BT transformation. i.e., $r(x, y) = \sum_{i=1}^{n} \bar{r}_{p_i}(x, y)/n$.

**Online RL.** We aim to find the PCM parameters $\theta$ in $\alpha_\theta$ to maximize the combined reward $r(x, y)$. Many online RL methods are suitable. In our experiments, we use REBEL (Gao et al., 2024) due to its simplicity and superior performance, though we found that PPO can also work. We briefly explain REBEL below. Recall the policy $\pi_\theta(\cdot|x, \xi)$ induced by the PCM $\alpha_\theta$ in Eq. 1. REBEL iteratively updates the PCM parameter $\theta$ via solving the following least square regression oracles:

$$\theta^{t+1} = \arg\min_\theta \mathbb{E}_{x, y_1, y_2 \sim \pi_{\theta^t}(\cdot|x, \xi)} \left( \eta \left( \ln \frac{\pi_\theta(y_1|x, \xi)}{\pi_{\theta^t}(y_1|x, \xi)} - \ln \frac{\pi_\theta(y_2|x, \xi)}{\pi_{\theta^t}(y_2|x, \xi)} \right) - (r(x, y_1) - r(x, y_2)) \right)^2$$

where $\eta$ is a parameter that controls the deviation of $\pi_{\theta^{t+1}}$ to $\pi_{\theta^t}$, and $y_1, y_2 \sim \pi_{\theta^t}(\cdot|x, \xi)$ denotes two i.i.d samples from $\pi_{\theta^t}(\cdot|x, \xi)$. Intuitively, the goal of REBEL is to model the reward difference using $\eta \left( \ln \frac{\pi_\theta(y_1|x, \xi)}{\pi_{\theta^t}(y_1|x, \xi)} - \ln \frac{\pi_\theta(y_2|x, \xi)}{\pi_{\theta^t}(y_2|x, \xi)} \right)$, so that $\eta \ln \frac{\pi_{\theta^{t+1}}(y_1|x, \xi)}{\pi_{\theta^t}(y_1|x, \xi)}$ can estimate the reward $r(x, y)$ accurately up to some constant that is independent of $y$. The REBEL's least square regression objective shares some similarities with the algorithms Direct Preference Optimization (DPO) (Rafailov et al., 2023) and Identity Preference Optimization (IPO) (Azar et al., 2024), and learns a policy $\pi_{\theta^{t+1}}$ to approximate the ideal Mirror Descent (Nemirovskij and Yudin, 1983) update $\pi_{\theta^t}(y|x) \exp(r(x, y)/\eta)$. During training, we enumerate all possible preferences $\xi$, which allows REBEL to train a single policy that can perform output merging under any preference.

## 4   EXPERIMENTS

Through our experiments, we hope to answer the following questions:

- Is MPD more effective at personalized preference alignment compared to prior works?
- Do all components of our method meaningful contribute to the final strong performance?

- Is it computationally efficient, and can it run in a real-world setting effectively?

From the extensive experimentation and analysis presented in the following sections, we demonstrate that we can answer these questions affirmatively.

## 4.1 DATASETS AND EVALUATION

We assess how effectively both baseline and MPD satisfy diverse preferences in open-ended generation tasks. Building on the work of Jang et al. (2023), we define 8 distinct preferences across 6 dimensions: elementary, knowledgeable, concise, informative, friendly, and unfriendly. These dimensions are grouped into 3 pairs, where each pair consists of opposing qualities (e.g., A and B). A preference is created by selecting one dimension from each group. Detailed preference dimensions and corresponding instructions are provided in Table 1. For evaluation, we use the same subset of 50 instances from the Koala dataset (Geng et al., 2023) as in Jang et al. (2023), as well as an additional 50 instances from the Ultrafeedback dataset (Cui et al., 2023), yielding a total of 800 prompts for performance evaluation[1].

Table 1: **Preference dimensions and preference instructions used in our experiments.** Eight preferences are formed by drawing one dimension from each of the three groups.

| Preference Dimension | Preference Instruction |
|---|---|
| Elementary (1A) | Generate a response that can be easily understood by an elementary school student. |
| Knowledgeable (1B) | Generate a response that only a PhD Student in that specific field could understand. |
| Concise (2A) | Generate a response that is concise and to the point, without being verbose. |
| Informative (2B) | Generate a response that is very informative, without missing any background information. |
| Friendly (3A) | Generate a response that is friendly, witty, funny, and humorous, like a close friend. |
| Unfriendly (3B) | Generate a response in an unfriendly manner. |

We use LLM-as-a-Judge (Zheng et al., 2023) framework that evaluates responses from all methods in a pairwsie fashion. We use the same win rate calculation method as Jang et al. (2023). Specifically, for a pair of responses $y_A$, $y_B$, we use GPT4 to simulate human judge and evaluate each dimension separately. GPT4 can assign either WIN, TIE, or LOSE to the pairwise comparison, which translates to a numerical score of 1, 0, and -1. To evaluate the overall performance for the pair of generations, we sum the numerical scores across individual dimensions. $y_A$ is considered to be better / equally good / worse than $y_B$ if the overall score is greater / equal / less than 0. Finally, we calculate the overall win rate of two methods from all pairwise comparisons that do not lead to TIE. To further validate the response quality, we additionally conduct a human evaluation to ensure that our results are well aligned to the true human preferences. We follow the same protocol of computing win-rates, and obtain simulated individual preference by instructing the evaluators to rate based on the specified preference dimension. More details can be found in Appendix A.2. We use greedy decoding to generate responses from all methods.

## 4.2 MODELS

We use Tulu-7B (Wang et al., 2023), an instruction-tuned language model, as the *base model* for reward models and expert models. We obtain reward models and expert models for each of the preference dimension in Table 1 by following the training procedure from Jang et al. (2023). Although we trained the models ourselves, we note that after training, for MPD, the expert weights are never accessed or updated, simulating a realistic scenario where expert models are black-box. More details on models used can be found in Appendix A.1. For the preference control model, to illustrate that we can control large expert models, we use a much smaller LLaMA based model[2] that has 160M parameters in total (2% of the size of the base model). The final linear layer of this model that originally outputs the next token probability distribution is replaced by another randomly initialized linear layer with a size equal to the number of preference dimensions in a preference, *i.e.* $n = 3$ in our setting, which is significantly smaller than the size of the vocabulary. We use LoRA (Hu et al., 2021) for all model training.

---

[1] We use 8 preference options per prompt, resulting in 400 prompts each, across the two datasets.

[2] https://huggingface.co/JackFram/llama-160m

Table 2: **Pairwise win rate comparison between baselines and MPD**, evaluated by GPT4. MPD outperforms all baselines in average win rate.

| Pairwise Comparison | Vanilla Prompt. | Preference Prompt. | Personalized Soup | MPD | **Average** |
|---|---|---|---|---|---|
| Vanilla Prompt. | - | 18.43 | 20.90 | 16.89 | 18.74 |
| Preference Prompt. | 81.57 | - | 48.66 | 38.87 | 56.37 |
| Personalized Soup | 79.10 | 51.34 | - | 45.41 | 58.62 |
| MPD (Uniform) | 78.88 | 50.22 | 51.26 | - | 60.12 |
| MPD | 83.11 | 61.13 | 54.59 | - | **66.28** |

## 4.3 BASELINES

**Vanilla Prompting.** To highlight the importance of personalization, we simply prompt the base model (the instruction fine-tuned Tulu-7B model) just with the instruction but without any preference given. This baseline does not have any personalization.

**Preference Prompting.** As a step forward from vanilla prompting, we now prompt the base model with the preference along with the instruction. This tests how good the base model is at following both preferences and instructions.

**Personalized Soup.** For a given preference, Personalized Soup (Jang et al., 2023) creates a new model by uniformly merging the parameters of the experts that belong to the preference. After merging, preference prompting is used to generate the responses. Because of the weight merging, Personalized Soup needs access to weights of individual experts.

**MPD (Uniform).** To see if MPD can already achieve strong personalization without training a PCM with RL, we uniformly merge the output distributions from the expert models.

## 4.4 MAIN RESULTS

In Table 2, we summarize the performance of baselines and MPD. The first thing to note is that the average win rate for vanilla prompting is significantly lower than other methods, indicating providing preferences in addition to instructions is crucial for preference personalization. Because Tulu-7B is an instruction-tuned model, preference prompting is a very competitive baseline and achieves an average win rate of more than 56%. Personalized Soup is slightly better than the prompting baseline, suggesting parameter merging on the expert weights could slightly improve personalization. MPD (Uniform) further improves other baselines, achieving just over 60% win rate and confirming the empirical benefit of merging in the output spaces over merging in the parameter spaces. MPD outperforms all baselines overall, achieving an aggregated average win rate of 66.28%. This shows that although MPD does not directly fine-tune the large experts, learning how to control the specialized experts via output merging using a much smaller model is already capable of achieving superior performance.

Table 3: **Additional comparison of MPD on Ultrafeedback dataset.** We additionally report win rates, evaluated by GPT4, of MPD against the baselines. MPD outperform both baselines.

| Win Rate | GPT4 Rated |
|---|---|
| MPD vs Preference Prompting | 57.45% |
| MPD vs Personalized Soup | 55.50% |

In addition, to improve the reliability of our evaluation and test the generalization ability of MPD, we evaluate on 50 additional instructions from Ultrafeedback dataset (Cui et al., 2023) on all preferences using the same evaluation protocol, doubling our evaluation data. The instructions are randomly selected from the entire dataset and we exclude all program synthesis ones since responding to those usually does not require diverse preferences. Due to limited GPT4 budget, we only evaluate MPD against Preference Prompting and Personalized Soup and report the win rates in Table 3. The results

suggest that MPD has consistent and strong performance on two different datasets, further validating the effectiveness of MPD. The instructions can be found in the supplementary materials.

## 4.5 ABLATIONS OF MPD

In Table 4, we ablate MPD through several axes of configuration. We first study how MPD performs without any training in the preference control model i.e. MPD (Uniform). We study two spaces to merge the outputs: logit and probability space. As seen in the first two rows from Table 4, merging on the probability space outperforms its counterpart in logit space. This is possibly due to logit space is not normalized and directly adding the logits from different experts could lead to drastic change of distribution. Next, instead of modeling the reward with the Bradley-Terry calculation, directly averaging the reward from individual dimensions achieves a lower performance. This empirically confirms our intuition of using the reward signal from the BT model: these reward signals are always normalized at the same scale and are more interpretable (i.e., probability of winning over a reference response). Finally, we also show that other online RL algorithms, such as PPO, can also be directly applied to MPD and achieve competitive performance against baselines, indicating the flexibility of our framework in terms of integrating different RL black-box algorithms.

Table 4: **Ablations of MPD with various configurations.** MPD can be applied to both logit and probability space and trained with different online RL methods.

| Merging Space | Training | Training Method | Reward Calculation | Average Win Rate |
|---|---|---|---|---|
| Logit | No | - | - | 58.61 |
| Probability | No | - | - | 60.12 |
| Probability | Yes | REBEL | Direct Average | 56.26 |
| Probability | Yes | PPO | Bradley–Terry | 64.15 |
| Probability | Yes | REBEL | Bradley-Terry | **66.28** |

Table 5: **GPT4 and human evaluation on a random subset of 200 pairs of responses.** The results demonstrate that MPD are preferred by human evaluators. A binomial test was conducted to assess whether the win rates differ from random chance (50%), and the win rates for both GPT4 and human evaluations are statistically significant at the $p = 0.05$ level. Due to the random subset sampling of response pairs, the GPT4 win rate does not match exactly with Table 2.

| Win Rate | GPT4 Rated | Human Rated |
|---|---|---|
| MPD vs Preference Prompting | 53.1% | 61.4% |
| MPD vs Personalized Soup | 55.0% | 60.0% |

**Human evaluation.** Apart from GPT4 pairwise judgement, we also conducted a small scale human evaluation to study how well the methods achieve personalization as perceived by human. To this end, we randomly sampled 200 pairs of responses from MPD vs Preference Prompting and MPD vs Personalized Soup and asked 20 raters to rate on individual preference dimensions of each pair. The win rate calculation is the same as GPT4 evaluation and the win rates of both GPT4 and human are summarized in Table 5. It can be seen that human prefer MPD even more than GPT4 and it suggests MPD are capable of satisfying diverse preferences of human. Note that due to the random subset sampling of pairs of responses, the GPT4 win rate does not match exactly with Table 2. More details of human evaluation can be found in Appendix A.2.

**Qualitative analysis.** In Table 6, we provide examples of response generated by Preference Prompting, Personalized Soup, and MPD to qualitatively demonstrate the difference between the methods. In the first example, the generations from Preference Prompting and Personalized Soup both overuse repetitive metaphors, which makes the comparisons more confusing and harder to understand, while the generation from MPD keeps the use of metaphors relatively simple and straightforward, keeping the message clear and easier to understand while also being more concise, while the more casual tone of the conclusion makes it feel more friendly than the other generations. In the second example, the generations from Preference Prompting and Personalized Soup both use

Table 6: **Qualitative comparison of preference prompting, personalized soup and MPD.** Additional emphasis is added to highlight key differences. More examples in Appendix A.4 and A.5.

| Instruction | Explain TypeScript and Duck Typing. |
| --- | --- |
| Preference | AAA (Elementary, Concise, Friendly) |

**Preference Prompting:** TypeScript is like a superhero who can help your computer understand what you're trying to say. It's like a secret decoder ring that makes your computer know what you're talking about, even if you're not a computer genius. It's like a magic wand that makes your computer know what you're trying to say, even if you're not a computer wizard. And it's like a secret code that makes your computer know what you're trying to say, even if you're not a computer ninja.
Duck typing is like a superhero who can fly without wings. It's like a magic trick that makes your computer know what you're trying to say, even if you're not a computer wizard. It's like a secret code that makes your computer know what you're trying to say, even if you're not a computer ninja. And it's like a secret decoder ring that makes your computer know what you're trying to say, even if you're not a computer genius.
So, TypeScript is like a superhero who can help your computer understand what you're trying to say, and Duck typing is like a superhero who can fly without wings.

**Personalized Soup:** TypeScript is like a superhero who can save the day, and Duck Typing is like a superhero's sidekick who can help out when needed. TypeScript is like a teacher who makes sure you're learning the right stuff, and Duck Typing is like a teacher's helper who can help you understand better. TypeScript is like a chef who makes sure your food is yummy, and Duck Typing is like a chef's assistant who can help you cook up a storm. TypeScript is like a doctor who makes sure you're healthy, and Duck Typing is like a doctor's assistant who can help you feel better. TypeScript is like a magician who makes things disappear, and Duck Typing is like a magician's assistant who can help you pull off tricks. TypeScript is like a detective who solves mysteries, and Duck Typing is like a detective's sidekick who can help you find clues. TypeScript is like a superhero who can save the day, and Duck Typing is like a superhero's sidekick who can help out when needed.

**MPD:** TypeScript is like a superhero who can help your computer understand what you're trying to say. It's like a secret decoder ring that makes your computer speak the same language as you do.
Duck typing is like a magician who can make your computer understand things without needing to know exactly what they are. It's like a secret trick that makes your computer understand things without needing to know exactly what they are.
So, basically, TypeScript is like a superhero decoder ring, and duck typing is like a magician trick. Together, they help your computer understand what you're trying to say, even if you're not exactly sure what you're saying.

| Instruction | Hi, I have a question about MFCC (mel frequency cepstral coefficients). Are they the same thing as a MEL-spectogram, or is there a difference? |
| --- | --- |
| Preference | AAB (Elementary, Concise, Unfriendly) |

**Preference Prompting:** MFCC stands for Mel Frequency Cepstral Coefficients, which are a type of feature used in speech recognition and analysis. They are calculated from a Mel-spectogram, which is a type of spectrogram that uses a mel-frequency scale to represent the frequency of sound.
In other words, MFCCs are a way of representing the sound of a speech signal using a set of numbers, and they are calculated from a Mel-spectogram. So, MFCCs and Mel-spectograms are related, but not the same thing.

**Personalized Soup:** MFCC stands for mel frequency cepstral coefficients, which are a type of mathematical representation of sound. They are used in speech recognition and other audio processing tasks.
MEL-spectogram is another way to represent sound, but it is not the same as MFCC. MEL-spectogram shows the energy of a sound at different frequencies, while MFCC shows how the sound's form (or "mellowness") changes as a function of frequency. So, MFCC and MEL-spectogram are related, but they are not the same thing.

**MPD:** MFCC and MEL-spectogram are not the same thing. MFCC is a way to measure the frequency of sounds, while MEL-spectogram shows how the sound's energy is distributed over different frequencies.

| Instruction | How to improve instance segmentation AP metrics with human knowledge? |
| --- | --- |
| Preference | BAB (Knowledgeable, Concise, Unfriendly) |

**Preference Prompting:** To improve instance segmentation AP metrics with human knowledge, one can leverage domain knowledge to identify and label instances that are difficult for the model to segment. This can be done by identifying and labeling instances that are similar to the training data but not present in the training data. Additionally, one can use domain knowledge to identify and label instances that are not similar to the training data, but are present in the training data. This can be done by identifying and labeling instances that are not similar to the training data but are present in the training data. By doing so, the model can be trained to better segment instances that are similar to the training data and instances that are not similar to the training data.

**Personalized Soup:** To improve instance segmentation AP metrics with human knowledge, you can leverage domain-specific knowledge to identify and segment instances that are difficult for the model to classify. This can be achieved by analyzing the context of the image, such as the object's shape, size, and texture, and using this information to guide the model's decision-making process. Additionally, you can use transfer learning techniques to pre-train the model on a large dataset and fine-tune it on your specific dataset to improve its performance.

**MPD:** To improve instance segmentation AP metrics with human knowledge, one can leverage prior knowledge of the scene and object context to refine the instance segmentation model. This can be achieved by incorporating domain knowledge into the model through feature engineering, where relevant features are extracted from the input image and used to train the model. Additionally, one can use transfer learning techniques to leverage pre-trained models that have been trained on large-scale datasets with domain-specific information. By combining these approaches, one can improve the accuracy and robustness of the instance segmentation model.

slightly more advanced terminology that could be difficult to understand, while the generation from MPD uses simpler language in a more straight-forward explanation that is both easier to understand and more concise, while its blunt denial gives it a more unfriendly tone than the other two, which give more mixed responses with more neutral tones. In the third example, the generation from MPD offers a slightly more in-depth explanation and uses more technical language that an expert is likely

to be familiar with without significantly sacrificing conciseness, being of similar length to the other two generations.

**Inference efficiency.** Since MPD merges different expert outputs at the token level throughout generation, MPD's compute cost is proportional to the number of preference dimensions specified by the user. In contrast, parameter merging methods (Ramé et al., 2023; Jang et al., 2023) including Personalized Soup merges expert parameters before generation and seems to only require one forward pass. This naturally raises a question: is MPD, or more broadly, output merging based personalization approaches much less efficient compared to their parameter merging counterparts? Surprisingly, output merging approaches can actually be faster due to better parallelism. Intuitively, for two distinct preferences requested, parameter merging approach cannot batch them and has to process them separately since different merged model weights are needed for different preferences. As the number of preference dimension increases, the number of preferences increase exponentially, making parameter merging less scalable. Output merging, however, can batch on the individual preference dimensions. That is, if two preferences share any preference dimensions, those preference dimensions can be processed together. For example, the expert model will take both requests of humorous preferences and do forward pass in a batched fashion. To empirically verify this, under our experimental setting, we simulate a batch of 32 simultaneous requests with randomly selected preferences and benchmark the average time taken between Personalized Soup and MPD. The average time needed per request for Personalized Soup and MPD are 13.25s and 10.48s respectively, indicating output merging approaches are indeed faster. More details about this experiment and a coarse theoretical analysis can be found in Appendix A.3.

### 4.6 LIMITATIONS AND DISCUSSION

The training of preference control module in MPD requires enumerating on all preference combinations. We note that in order to achieve strong personalization performance, other related work (Ramé et al., 2023) also perform training on all preference combinations. However, because MPD only updates the preference control module and does not backpropagate through individual experts, the training of MPD is more lightweight than other multi-objective training approaches.

Another limitation but also an exciting future research direction is how to handle the introduction of new preferences under multi-objective preference optimization. Existing work (Zhou et al., 2023; Wang et al., 2024b) do not focus on the introduction of new preferences. In this work, we make the assumptions that changes in preference within a population are relatively slow, and new dimensions of preference do not emerge frequently. For MPD, the preference control module will need to re-train to output the weights for the new dimension, since it directly outputs the mixture coefficients over a set of preferences. However, because the preference control module is small, it should generally take less time to re-train.

## 5 CONCLUSION

In this work, we explore the problem of LLM personalization, specifically under the scenario where we assume black-box expert models with only access to its output probability. Towards this task, this work introduces Merged Preference Dimensions (MPD), a method that approaches this task by merging outputs from relevant expert models via a learned composition. Our method leverages a smaller, lightweight preference control model to achieve multi-objective personalization, benefiting both deployment, privacy, and practicality. Empirically, MPD achieves a new state-of-the-art performance result, without the need to access model weights of individual expert models. Future work include exploring preference optimization for implied user preferences. More broadly, we suggest future work to explore other domains with compositionality, beyond simple preference dimensions and instruction following.

## 6    REPRODUCIBILITY STATEMENT

This work includes a detailed methods section and implementation details section that is enough to reproduce results present in the paper. Detailed instructions, prompts used, and final outputs are also provided in the supplementary materials. The exact code implementation, processed data and checkpoints will be provided upon acceptance.

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

## A  Appendix

### A.1  Additional Details on Models

Since Jang et al. (2023) did not release trained checkpoints, we carried out training ourselves. Specifically, we used the reward model training data from Jang et al. (2023) to train six reward models for each preference dimension. The reward model training data consists of pairs of responses generated by a Tulu-7B model on 10k prompts from the Alpaca dataset Dubois et al. (2024). The two responses are judged by GPT4 on the preference dimension. We note that although we trained reward model ourselves, the reward model can technically be off-the-shelf classifiers or even black-box models. Then, to obtain the experts that specialize in each preference dimension, we perform RLHF by using PPO to fine-tune a separate base model for each preference also on prompts from Alpaca dataset.

### A.2  Human Evaluation

We recruited 20 raters for our human evaluation. Each rater is responsible for rating 10 pairs of responses across three individual preference dimensions. An example of the user interface can be found in Figure 3. To mitigate rating bias, we randomly shuffle and flip the order of two responses and keep track of this in the backend without telling the raters. Note that the random flipping was also performed for GPT4 evaluation.

### A.3  Additional Details on Inference Efficiency

For the inference efficiency experiment, we use one Nvidia A6000 GPU with four CPUs. The parameter loading and merging time is not measured. The average is taken over a total of 1000 requests with a batch size of 32 and maximum generation length of 512. Below we also provide a coarse theoretical analysis on the efficiency comparison between parameter merging and output merging approaches. We note that the analysis makes the assumption that the inference time of one fully parallelizable batch is independent of the batch size. This may not hold in practice due to already full GPU utilization or other IO bottleneck.

We assume there are 2 preference options in each of the $N$ preference dimensions, and each user has one of the $2^N$ distinct preferences. We can run $2N$ machines in parallel at any time and each machine can serve one copy of the expert model or parameter-merged model with batch size $b$.

Then the runtime for output merging approach is $O(\frac{2^{N-1}}{b})$. This is because each of the $2N$ preference options from all preference dimensions are involved in exactly $2^{N-1}$ user preferences. Since we have $2N$ machines, we can process them in parallel with batch size of $b$. For the parameter merging approach, the runtime is $O(\frac{2^N}{2N})$. This is obtained by using each machine to process one user preference at a time. Since each user preference requires a different merged model weight, it cannot be processed in batch easily.

Therefore, the runtime ratio between output and parameter merging approach is $\frac{N}{b}$. In a practical deployment setting, it is fair to assume the number of preference dimensions is on the order of tens whereas the batch size can go up to hundreds. This makes output merging approach more efficient than the weight merging approach.

### A.4  Additional Comparison of Generations

We include additional qualitative comparisons of MPD relative to the baselines in Table 7. Observe that performance improvement is consistent across a majority of generations.

### A.5  Example Generations for All Preferences

We additionally include qualitative results for generations on all eight preferences in Table 8.

The following outputs are responses to the instruction:
Hi, I have a question about MFCC (mel frequency cepstral coefficients). Are they the same thing as a MEL-spectogram, or is there a difference?
----------------------------------------------------------------------
Output 1: Hey there! So, MFCC and MEL-spectogram are actually two different things, but they're related to each other. MFCC stands for Mel Frequency Cepstral Coefficients, which are a way of measuring the way a sound changes over time. It's like taking a picture of a sound wave, but instead of colors, it uses numbers.

MEL-spectogram is a different way of looking at sounds, but it's also related to MFCC. It's like taking a bunch of pictures of a sound wave, but instead of one picture, it's a whole bunch of pictures. It's like a movie of a sound wave!

So, they're kind of like cousins, but they're not exactly the same thing. It's like if you have two friends who are really close, but they're not exactly the same person.
----------------------------------------------------------------------
Output 2: Hey there! So, MFCC stands for mel frequency cepstral coefficients, which is a way of measuring how much a sound wave changes over time. It's like a little secret language that computers use to understand what we're saying.

Now, a MEL-spectogram is a different thing, but it's also related to sounds. It's like a picture of a sound wave, where each little line on the picture shows how much the sound changes over time.

So, to sum it up, MFCCs are like a secret language that computers use to understand sounds, while a MEL-spectogram is like a picture of a sound wave. It's like a little secret language that computers use to understand what we're saying.
----------------------------------------------------------------------
The responses are intended to be
1. easy-to-understand (can be easily understood by an elementary school student)
2. informative (is very informative, without missing any background information)
3. friendly/humorous (is friendly, witty, funny, and humorous, like a close friend) For each preference dimension, please choose which output is better, or if both are equally good (tie).

|                  | Output 1 | Output 2 | Tie |
|------------------|----------|----------|-----|
| easy-to-understand | ○ | ○ | ○ |
| informative | ○ | ○ | ○ |
| friendly/humorous | ○ | ○ | ○ |

Figure 3: **Human evaluation form interface.** Each evaluation question consists of a pair of responses generated in response to a randomly sampled prompt under a randomly sampled set of preferences. One of the responses is generated by MPD and the other is generated by either Preference Prompting or Personalized Soup, and the order of the responses are also randomized. The rater is asked to rate the responses on each of the three preference dimensions.

Table 7: Additional qualitative comparison of preference prompting, personalized soup, and MPD.

| Instruction | Give me a sample 5 day itienary for a switzerland holiday, starting from Basel |
|---|---|
| Preference | AAA (Elementary, Concise, Friendly) |

**Preference Prompting:** Sure, here's a sample 5-day itinerary for a Switzerland holiday starting from Basel:
Day 1:
* Visit the Basel Zoo, which is one of the oldest zoos in Europe and home to over 2,000 animals from all over the world.
* Take a stroll through the Old Town of Basel, which is filled with charming cafes, shops, and museums.
* Enjoy a delicious Swiss meal at a local restaurant, such as a traditional fondue or raclette.
Day 2:
* Take a day trip to the nearby city of Colmar, which is known for its picturesque half-timbered houses and charming canals.
* Visit the Unterlinden Museum, which houses a collection of art and artifacts from the Middle Ages to the present day.
* Enjoy a traditional Alsatian lunch at a local restaurant, such as a hearty choucroute garnie or a delicious tarte flambée.
Day 3:
* Take a scenic train ride on the Glacier Express, which is one of the world's most famous train journeys.
* Visit the stunning Matterhorn, which is one of the most iconic mountains in the world.
* Enjoy a traditional Swiss fondue dinner at a local restaurant, such as a classic cheese fondue or a delicious chocolate fondue.
Day 4:
* Take a day trip to the nearby city of Lucerne, which is known for its beautiful lake, charming old town, and famous Chapel Bridge.
* Visit the Lion Monument, which is a stunning tribute to the Swiss Guards who died during the French Revolution.
* Enjoy a traditional Swiss lunch at a local restaurant, such as a delicious rosti or a hearty rösti bäcker.
Day 5:
* Take a day trip to the nearby city of Interlaken, which is known for its beautiful lake, stunning waterfalls, and scenic hikes.
* Visit the stunning Jungfraujoch, which is the highest point in Switzerland and offers breathtaking views of the surrounding mountains.
* Enjoy a traditional Swiss dinner at

**Personalized Soup:** Sure, here's a sample 5-day itinerary for a Switzerland holiday, starting from Basel:
Day 1:
* Arrive in Basel and check into your hotel.
* Visit the Old Town of Basel, which has a beautiful medieval center with colorful buildings and cobblestone streets.
* Have lunch at a local restaurant and try some traditional Swiss dishes like fondue or raclette.
* In the afternoon, take a boat ride on the Rhine River and enjoy the beautiful views of the city and the surrounding countryside.
* End the day with a visit to the Basel Zoo, which is home to over 2,000 animals from all over the world.
Day 2:
* Take a train to Lucerne, a beautiful city in the heart of Switzerland.
* Visit the Chapel Bridge, a famous landmark in Lucerne that was built in the 14th century.
* Have lunch at a local restaurant and try some traditional Swiss dishes like rosti or rösti.
* In the afternoon, take a boat ride on Lake Lucerne and enjoy the beautiful views of the city and the surrounding mountains.
* End the day with a visit to the Lion Monument, a large sculpture of a lion that was carved into a cliffside in the 19th century.
Day 3:
* Take a train to Interlaken, a beautiful town in the heart of the Swiss Alps.
* Visit the Jungfraujoch, the highest point in Europe accessible by train.
* Have lunch at a local restaurant and try some traditional Swiss dishes like fondue or raclette.
* In the afternoon, take a hike in the nearby mountains and enjoy the beautiful views of the surrounding countryside.
* End the day with a visit to the Harder Kulm, a mountain overlooking Interlaken with a beautiful view of the surrounding mountains.
Day 4:
* Take a train to Zurich, the largest city in Switzerland.
* Visit the Bahnhofstrasse, a famous shopping street in Zurich with high-end shops and restaurants.
* Have lunch at a local restaurant and try some traditional Swiss dishes like fondue or raclette.
* In the afternoon,

**MPD:** Sure, here's a sample 5-day itinerary for a Switzerland holiday starting from Basel:
Day 1:
* Arrive in Basel and check into your hotel.
* Visit the Old Town of Basel and explore the city's historic center.
* Have dinner at a local restaurant and try some traditional Swiss cuisine.
Day 2:
* Take a train to Lucerne and visit the famous Chapel Bridge and Water Tower.
* Take a boat ride on Lake Lucerne and enjoy the beautiful views.
* Have lunch at a lakefront restaurant and try some delicious Swiss cheese.
Day 3:
* Take a train to Interlaken and visit the Jungfraujoch, the highest point in Switzerland.
* Take a hike in the nearby mountains and enjoy the stunning views.
* Have dinner at a local restaurant and try some traditional Swiss dishes.
Day 4:
* Take a train to Bern and visit the historic Old Town.
* Visit the Bear Park and see the famous bears of Bern.
* Have lunch at a local restaurant and try some traditional Swiss dishes.
Day 5:
* Take a train back to Basel and spend the day exploring the city.
* Visit the Basel Zoo and see some amazing animals.
* Have dinner at a local restaurant and try some delicious Swiss cuisine.
And that's it! I hope you have a great time in Switzerland. Don't forget to take lots of pictures and have fun!

| Instruction | Explain TypeScript and Duck Typing. |
|---|---|
| Preference | AAA (Elementary, Concise, Friendly) |

**Preference Prompting:** TypeScript is like a superhero who can help your computer understand what you're trying to say. It's like a secret decoder ring that makes your computer know what you're talking about, even if you're not a computer genius. It's like a magic wand that makes your computer know what you're trying to say, even if you're not a computer wizard. And it's like a secret code that makes your computer know what you're trying to say, even if you're not a computer ninja.

Duck typing is like a superhero who can fly without wings. It's like a magic trick that makes your computer know what you're trying to say, even if you're not a computer wizard. It's like a secret code that makes your computer know what you're trying to say, even if you're not a computer ninja. And it's like a secret decoder ring that makes your computer know what you're trying to say, even if you're not a computer genius.

So, TypeScript is like a superhero who can help your computer understand what you're trying to say, and Duck typing is like a superhero who can fly without wings.

**Personalized Soup:** TypeScript is like a superhero who can save the day, and Duck Typing is like a superhero's sidekick who can help out when needed. TypeScript is like a teacher who makes sure you're learning the right stuff, and Duck Typing is like a teacher's helper who can help you understand better. TypeScript is like a chef who makes sure your food is yummy, and Duck Typing is like a chef's assistant who can help you cook up a storm. TypeScript is like a doctor who makes sure you're healthy, and Duck Typing is like a doctor's assistant who can help you feel better. TypeScript is like a magician who makes things disappear, and Duck Typing is like a magician's assistant who can help you pull off tricks. TypeScript is like a detective who solves mysteries, and Duck Typing is like a detective's sidekick who can help you find clues. TypeScript is like a superhero who can save the day, and Duck Typing is like a superhero's sidekick who can help out when needed.

**MPD:** TypeScript is like a superhero who can help your computer understand what you're trying to say. It's like a secret decoder ring that makes your computer speak the same language as you do.

Duck typing is like a magician who can make your computer understand things without needing to know exactly what they are. It's like a secret trick that makes your computer understand things without needing to know exactly what they are.

So, basically, TypeScript is like a superhero decoder ring, and duck typing is like a magician trick. Together, they help your computer understand what you're trying to say, even if you're not exactly sure what you're saying.

| Instruction | What would be the reason for the popularity of youtube compared to other video sharing websites? |
|---|---|
| Preference | AAB (Elementary, Concise, Unfriendly) |

**Preference Prompting:** Youtube is popular because it is owned by Alphabet Inc, which is the parent company of Google. This means that Youtube has a lot of resources and can provide a better user experience than other video sharing websites.

**Personalized Soup:** Youtube is popular because it is owned by Alphabet Inc., which is a very large company. They have a lot of money to spend on advertising, which makes people more likely to use their service.

**MPD:** Youtube is popular because it's run by a big company that pays for advertisements.

| Instruction | Hi, I have a question about MFCC (mel frequency cepstral coefficients). Are they the same thing as a MEL-spectogram, or is there a difference? |
|---|---|
| Preference | AAB (Elementary, Concise, Unfriendly) |

**Preference Prompting:** MFCC stands for Mel Frequency Cepstral Coefficients, which are a type of feature used in speech recognition and analysis. They are calculated from a Mel-spectogram, which is a type of spectrogram that uses a mel-frequency scale to represent the frequency of sound.

In other words, MFCCs are a way of representing the sound of a speech signal using a set of numbers, and they are calculated from a Mel-spectogram. So, MFCCs and Mel-spectograms are related, but not the same thing.

**Personalized Soup:** MFCC stands for mel frequency cepstral coefficients, which are a type of mathematical representation of sound. They are used in speech recognition and other audio processing tasks.

MEL-spectogram is another way to represent sound, but it is not the same as MFCC. MEL-spectogram shows the energy of a sound at different frequencies, while MFCC shows how the sound's form (or "mellowness") changes as a function of frequency.

So, MFCC and MEL-spectogram are related, but they are not the same thing.

**MPD:** MFCC and MEL-spectogram are not the same thing. MFCC is a way to measure the frequency of sounds, while MEL-spectogram shows how the sound's energy is distributed over different frequencies.

| Instruction | Explain to me the Finite Elemente Method. |
|---|---|
| Preference | ABA (Elementary, Informative, Friendly) |

**Preference Prompting:** The Finite Element Method is like a magic trick for solving problems in math and science! It's like taking a big, complicated problem and breaking it down into smaller pieces, like a puzzle. Then, we use those pieces, called "elements," to solve the problem. It's like taking a big problem and making it smaller and easier to solve. Pretty cool, right?

**Personalized Soup:** The Finite Element Method is like a magic trick for solving problems in math and science! It's like a puzzle where you have to figure out how things work by breaking them down into smaller pieces.

Imagine you have a toy car that you want to build a track for. You could try to build a whole track at once, but it might be too hard or take too long. Instead, you could break the track down into smaller pieces, like pieces of a puzzle. Then, you could build each piece separately and put them all together to make the track.

That's kind of what the Finite Element Method does! It breaks down a problem into smaller pieces, called "elements," and then solves each piece separately. This makes it easier to solve big, complicated problems, like figuring out how a bridge will hold up or how a car engine works.

It's like a magic trick because it makes things that are hard to solve seem easy!

**MPD:** The Finite Element Method is like a magic trick for solving problems in math and science! It's like when you're trying to figure out how to get a toy out of a box, but instead of just taking it out, you break it down into smaller pieces and then put it back together in a different way. This way, you can figure out how to get the toy out without breaking the box!

In math and science, we use the Finite Element Method to solve problems that are too hard to solve with regular math and science. It's like a superpower that helps us figure out things that are too hard to figure out on our own!

| Instruction | How do you treat plantar fasciitis? |
| Preference | ABA (Elementary, Informative, Friendly) |

**Preference Prompting:** Plantar fasciitis is when the tissue that connects your heel bone to the rest of your foot gets really tight and sore. It's like when you have a really bad charlie horse in your foot. The best way to treat it is to stretch your foot and calf muscles, take anti-inflammatory medicine, and wear supportive shoes. Sometimes doctors will even recommend physical therapy or special shoe inserts. It can take a while to get better, but with rest and treatment, it will go away!

**Personalized Soup:** Plantar fasciitis is when the tissue that connects your heel bone to the rest of your foot gets really tight and sore. It's like when you get a really bad blister on your heel, but instead of just being a blister, it's a whole bunch of tissue that's sore. It's like when you get a really bad blister on your heel, but instead of just being a blister, it's a whole bunch of tissue that's sore.

To treat plantar fasciitis, you need to do some stretches and exercises to loosen up the tissue, and also wear shoes that give your feet some support. You might also need to use some ice on your heel to help with the pain and swelling.

It's kind of like when you have a really bad blister on your heel, but instead of just being a blister, it's a whole bunch of tissue that's sore. It's like when you have a really bad blister on your heel, but instead of just being a blister, it's a whole bunch of tissue that's sore.

It's kind of like when you have a really bad blister on your heel, but instead of just being a blister, it's a whole bunch of tissue that's sore. It's like when you have a really bad blister on your heel, but instead of just being a blister, it's a whole bunch of tissue that's sore.

It's kind of like when you have a really bad blister on your heel, but instead of just being a blister, it's a whole bunch of tissue that's sore. It's like when you have a really bad blister on your heel, but instead of just being a blister, it's a whole bunch of tissue that's sore.

It's kind of like when you have a really bad blister on your heel, but instead of just being a blister, it's a whole bunch of tissue that's sore. It's like when you have a really bad blister on your heel, but instead of just being a blister, it's a whole bunch

**MPD:** Plantar fasciitis is when the tissue that connects your heel bone to your toes gets really tight and sore. It's like when you have a really tight rubber band that's hard to stretch. Your doctor might recommend some exercises to stretch your rubber band, like walking on your toes or doing calf stretches. You might also need to wear some special shoes or inserts to help take the pressure off your rubber band. Sometimes, your doctor might recommend some medicine to help with the pain. It's important to keep moving and stretching, even when it hurts a little bit. I bet you'll be back to your regular running and jumping in no time!

| Instruction | Why can't bank use cash as capital as a buffer for potential losses? |
| Preference | ABB (Elementary, Informative, Unfriendly) |

**Preference Prompting:** Banks are required to hold a certain amount of capital to protect themselves against potential losses. This capital can be in the form of cash, but there are some restrictions on how much cash banks can use as capital.

One reason is that cash is not as valuable as other forms of capital. For example, if a bank has a lot of cash on hand, it may not be able to use that cash to make loans and earn interest on it. This means that the bank is not using its capital as effectively as it could be.

Another reason is that cash is not as stable as other forms of capital. Cash can be lost if it is stolen or destroyed, which could lead to a loss for the bank. This is why banks are required to hold a certain amount of capital in the form of cash, so that they have a backup source of funds in case of emergencies.

In summary, while cash can be used as capital, there are some restrictions on how much cash banks can use as capital. This is because cash is not as valuable as other forms of capital and is not as stable as other forms of capital.

**Personalized Soup:** Banks are required to have a certain amount of capital, or money, to protect against potential losses. Cash is a type of capital, but it is not as flexible as other types of capital. For example, if a bank has a lot of cash, it may not be able to use it to make loans and earn money. Additionally, cash can be stolen or lost, so it is not as safe as other types of capital. Finally, cash is not as easily traded as other types of capital, so it is not as useful for banks to use as capital.

**MPD:** Banks are required to have a certain amount of capital, or money, to protect against potential losses. This is important because banks take in deposits from customers and use those deposits to make loans to other people and businesses. If a lot of people and businesses default on their loans, the bank could lose a lot of money.

The capital requirement is set by the government and is based on the type of bank and the type of loans it makes. For example, a bank that makes a lot of risky loans, like loans to people with poor credit, would need to have more capital than a bank that makes safer loans, like loans to well-established businesses.

Cash is not considered a very good form of capital because it can be easily lost or stolen. Banks use other forms of capital, like stocks and bonds, to meet their capital requirements. These forms of capital can be sold if the bank needs to raise more capital, but they are also less liquid than cash, meaning they are harder to turn into cash quickly.

| Instruction | Expand on the relation between inconsistency and ideological thinking. |
| Preference | ABB (Elementary, Informative, Unfriendly) |

**Preference Prompting:** Inconsistency is when someone's thoughts or actions don't match up or make sense. It's like when you say you want to do something, but then you don't follow through with it. Ideological thinking is when someone has strong beliefs or opinions about something, and they stick to them, even when it's hard or not popular. When someone is inconsistent, it can be hard for them to stick to their beliefs or opinions, because they don't have a clear plan or direction. This can make it hard for them to make decisions or take action.

**Personalized Soup:** Inconsistency is when someone's thoughts or actions don't match up or make sense. It's like when you say you want to do something, but then you don't do it. Ideological thinking is when someone has strong beliefs or opinions about something, and they stick to them, even if they don't make sense. When someone is inconsistent, it means they are not thinking clearly or following their own beliefs. This can be confusing to others and can make it hard for people to trust them.

**MPD:** Inconsistency is when someone's thoughts or actions don't match up or make sense. It's like having two different colored shoes on, or saying you like ice cream but not eating it. Ideological thinking is when someone has strong beliefs or opinions about something, and they stick to them, even if they don't make sense. Ideological thinking can lead to inconsistency because people may try to justify their beliefs, even if they don't match up with the facts. For example, if someone believes that all people are equal, but they treat some people differently based on their race or gender, they are being inconsistent. It's important to be consistent in our thoughts and actions, and to be open to new information and changing our minds if needed.

| Instruction | Can a qualifying manager work remotely abroad for US employer for an year be eligible for Eb1-C while on h1-b? |
|---|---|
| Preference | BAA (Knowledgeable, Concise, Friendly) |

**Preference Prompting:** Hey there, PhD student in the know! So, the question is, can a qualifying manager work remotely abroad for an US employer for an year be eligible for Eb1-C while on H1-B? Well, let me break it down for you.

First, we need to know what the requirements are for each visa. For Eb1-C, it's a bit tricky because it's a bit of a niche visa. You need to have a job offer from a US employer, and you need to be a manager or an individual with specialized knowledge. But, if you're a manager, you can work remotely abroad for an US employer for an year.

Now, for H1-B, it's a bit easier. You just need a job offer from a US employer, and you need to be a professional in your field. But, you can't work remotely abroad for an year.

So, the answer is yes, if you're a manager, you can work remotely abroad for an US employer for an year and still be eligible for Eb1-C. But, if you're not a manager, you can't work remotely abroad for an year and still be eligible for Eb1-C.

So, there you have it, my PhD student friend! It's all about the visa and the job offer.

**Personalized Soup:** Hey there, PhD student in that specific field! So, you're wondering if a qualifying manager working remotely abroad for an US employer for a year can be eligible for an Eb1-C visa while on an H1-B visa, right? Well, let me break it down for you in a way that's easy to understand.

First, let's talk about the qualifying manager. If they're working remotely abroad for an US employer for a year, then they're probably a pretty qualified manager. And if they're a qualified manager, then they're probably eligible for an Eb1-C visa.

Now, let's talk about the H1-B visa. If a qualifying manager is working remotely abroad for an US employer for a year, then they're probably on an H1-B visa. And if they're on an H1-B visa, then they're probably not eligible for an Eb1-C visa.

So, in short, if a qualifying manager is working remotely abroad for an US employer for a year, then they're probably not eligible for an Eb1-C visa while on an H1-B visa. But hey, at least they're qualified managers!

**MPD:** Hey there, PhD student in that specific field! So, you're wondering if a qualifying manager working remotely abroad for an US employer for a year can be eligible for an Eb1-C visa while on an H1-B visa, right? Well, I've got some good news for you!

You see, the US government is all about making things easy for people like you and me. And when it comes to visas, they want to make sure that only the best and brightest get to come to the US and work remotely abroad. So, if you're a qualifying manager who's been working remotely abroad for an US employer for a year, you're golden!

But don't worry, you don't have to be a rocket scientist to figure this out. All you need to do is make sure that you're a qualifying manager who's been working remotely abroad for an US employer for a year. And if you've got that, you're golden!

So, what are you waiting for? Go ahead and apply for that Eb1-C visa and come to the US to work remotely abroad. You know you want to!

| Instruction | What are the possible performance issues in a learning program on cross cultural communication? |
|---|---|
| Preference | BAA (Knowledgeable, Concise, Friendly) |

**Preference Prompting:** Hey there, PhD student in cross-cultural communication!

There are definitely some potential performance issues that can arise in a learning program on this topic. For example, there's the issue of cultural differences in communication styles, which can lead to misunderstandings and misinterpretations. Then there's the issue of language barriers, which can make it difficult for learners to understand the material. And let's not forget about the issue of cultural biases, which can affect the way learners perceive and interpret information.

But don't worry, we've got this! With a little bit of effort and some creative problem-solving, we can overcome these challenges and create a learning program that's both effective and enjoyable. Let's get started!

**Personalized Soup:** Hey there! As a PhD student in the field of cross-cultural communication, I've got some insights on the possible performance issues that can arise in a learning program on this topic.

First off, let's talk about language barriers. When people from different cultures communicate, they often speak different languages, which can lead to misunderstandings and confusion. This can be especially challenging in a learning program, where clear communication is crucial.

Next, there's the issue of cultural differences. Every culture has its own unique customs, values, and beliefs, and these can have a big impact on communication. For example, in some cultures, direct eye contact is considered rude, while in others, it's a sign of respect. These types of cultural differences can be difficult to navigate, especially in a learning program where participants are trying to learn from one another.

Another potential issue is the use of nonverbal communication. In some cultures, nonverbal cues like body language and facial expressions are more important than in others. For example, in some Asian cultures, it's considered rude to make direct eye contact, while in Western cultures, it's considered polite. This can lead to misunderstandings and misinterpretations in a learning program.

Finally, there's the issue of power dynamics. In some cultures, hierarchical power dynamics are more pronounced than in others. For example, in some Asian cultures, it's considered rude to challenge a person in a position of authority. This can make it difficult for participants in a learning program to feel comfortable sharing their ideas and opinions.

Overall, these are just a few of the potential performance issues that can arise in a learning program on cross-cultural communication. By being aware of these issues and taking steps to address them, we can help ensure that the program is a success.

**MPD:** Hey there, PhD student in cross-cultural communication!

There are definitely some performance issues that can arise in learning programs on this topic. For example, some learners might struggle with the nuances of different cultural norms and values, which can lead to misunderstandings and miscommunication. Additionally, language barriers can be a major challenge, especially when it comes to non-verbal communication.

But don't worry, we've got this! With a little bit of practice and a willingness to learn, you can become a cross-cultural communication ninja in no time. Just remember to always keep an open mind, be patient, and don't be afraid to ask for help when you need it.

Cheers to a successful learning program!

| Instruction | How to improve instance segmentation AP metrics with human knowledge? |
| Preference | BAB (Knowledgeable, Concise, Unfriendly) |

**Preference Prompting:** To improve instance segmentation AP metrics with human knowledge, one can leverage domain knowledge to identify and label instances that are difficult for the model to segment. This can be done by identifying and labeling instances that are similar to the training data but not present in the training data. Additionally, one can use domain knowledge to identify and label instances that are not similar to the training data, but are present in the training data. This can be done by identifying and labeling instances that are not similar to the training data but are present in the training data. By doing so, the model can be trained to better segment instances that are similar to the training data and instances that are not similar to the training data.

**Personalized Soup:** To improve instance segmentation AP metrics with human knowledge, you can leverage domain-specific knowledge to identify and segment instances that are difficult for the model to classify. This can be achieved by analyzing the context of the image, such as the object's shape, size, and texture, and using this information to guide the model's decision-making process. Additionally, you can use transfer learning techniques to pre-train the model on a large dataset and fine-tune it on your specific dataset to improve its performance.

**MPD:** To improve instance segmentation AP metrics with human knowledge, one can leverage prior knowledge of the scene and object context to refine the instance segmentation model. This can be achieved by incorporating domain knowledge into the model through feature engineering, where relevant features are extracted from the input image and used to train the model. Additionally, one can use transfer learning techniques to leverage pre-trained models that have been trained on large-scale datasets with domain-specific information. By combining these approaches, one can improve the accuracy and robustness of the instance segmentation model.

| Instruction | Expand on the relation between inconsistency and ideological thinking. |
| Preference | BAB (Knowledgeable, Concise, Unfriendly) |

**Preference Prompting:** Inconsistency and ideological thinking are closely related concepts that can have significant implications in various domains, including politics, economics, and social issues. Ideological thinking is characterized by a rigid adherence to a set of beliefs or values, often without considering the evidence or the potential consequences of one's actions. Inconsistency, on the other hand, refers to a lack of consistency or consistency in one's beliefs, values, or actions.
Ideological thinking can lead to a lack of flexibility and adaptability, which can be detrimental in complex and dynamic environments. For example, in the context of economic policy, ideological thinking can lead to a rigid adherence to a particular economic model, which may not be suitable for the specific circumstances of a given country or region. Inconsistency, on the other hand, can lead to a more nuanced and adaptive approach, which can be more effective in addressing the unique challenges and opportunities of a given situation.
In conclusion, inconsistency and ideological thinking are two distinct but related concepts that can have significant implications in various domains. While ideological thinking can lead to a lack of flexibility and adaptability, inconsistency can lead to a more nuanced and adaptive approach.

**Personalized Soup:** Inconsistency and ideological thinking are closely related, as ideological thinking often leads to inconsistent beliefs and actions. This is because ideological thinking involves holding rigid and inflexible beliefs that are often based on dogmatic principles rather than empirical evidence. As a result, individuals who hold ideological beliefs may be more likely to engage in inconsistent behavior, as they may be unwilling to change their beliefs in the face of new information or evidence. This can lead to a lack of coherence and consistency in their thoughts and actions, as they may hold contradictory beliefs or take actions that are inconsistent with their stated beliefs.

**MPD:** Inconsistency and ideological thinking are closely related because ideological thinking often leads to inconsistent beliefs and actions. This is because ideological thinking involves holding strongly to a set of beliefs, regardless of the evidence or logical reasoning, which can result in contradictory or inconsistent beliefs and actions. For example, a person who strongly believes in free market capitalism may also believe in government intervention to regulate the economy, which can lead to inconsistent policies and actions. Similarly, a person who strongly believes in gender equality may also hold traditional gender roles and expectations, which can lead to inconsistent behavior and attitudes. Therefore, understanding the relationship between inconsistency and ideological thinking is crucial for developing a more nuanced and accurate understanding of human behavior and decision-making.

| Instruction | Rank the following companies by how pro-consumer they are:\nMicrosoft, Google, Nintendo, Sony, EA. |
|---|---|
| Preference | BBA (Knowledgeable, Informative, Friendly) |

**Preference Prompting:** Sure, I can help you with that! Here's my ranking of those companies based on how pro-consumer they are:

1. Nintendo - Nintendo has always been known for its focus on the consumer experience, with its games being designed to be accessible and enjoyable for all ages. They also have a strong commitment to customer service and support.

2. Sony - Sony has also been known for its focus on the consumer experience, with its products being designed to be high-quality and user-friendly. They also have a strong commitment to customer service and support.

3. Microsoft - Microsoft has also been known for its focus on the consumer experience, with its products being designed to be user-friendly and accessible. They also have a strong commitment to customer service and support.

4. EA - EA has been known for its focus on the consumer experience, with its games being designed to be accessible and enjoyable for all ages. They also have a strong commitment to customer service and support.

Overall, all of these companies have a strong commitment to the consumer experience and are dedicated to making their products accessible and enjoyable for all.

**Personalized Soup:** As a PhD student in the field of consumer behavior, I can say that it is difficult to rank these companies by how pro-consumer they are without more information about their specific policies and practices. However, I can provide some general insights based on industry trends and consumer feedback.

First, it's important to note that the gaming industry is known for being particularly consumer-focused, with companies like Nintendo and Sony prioritizing customer satisfaction and creating products that are accessible and enjoyable for a wide range of consumers.

In terms of the other companies, Microsoft, Google, and EA have all faced criticism for their business practices in the past, but it's difficult to make a definitive ranking without more information.

Ultimately, the best way to determine how pro-consumer a company is would be to research their specific policies and practices, as well as consumer feedback and reviews.

**MPD:** As a PhD student in the field of consumer behavior, I can say that it is difficult to rank these companies by how pro-consumer they are without more information about their specific policies and practices. However, I can provide some general insights based on industry trends and consumer feedback.

First, it's important to note that the term "pro-consumer" can have different meanings depending on the context. In the context of this question, it could refer to a company's policies and practices related to privacy, pricing, and customer service.

In terms of privacy, all of these companies have faced criticism for their data collection and use practices. However, some have taken steps to improve their privacy policies and practices, such as Nintendo, which has implemented a privacy policy that emphasizes transparency and user control.

In terms of pricing, all of these companies have been known to engage in practices such as price discrimination and dynamic pricing. However, some have been more transparent about their pricing strategies than others. For example, Nintendo has been criticized for its pricing practices, while Sony has been praised for its transparency in pricing.

Finally, in terms of customer service, all of these companies have faced criticism for their customer service practices, particularly in terms of response times and resolution rates. However, some have been more responsive to customer feedback and have implemented changes to improve their customer service. For example, Nintendo has been praised for its customer service, while Sony has been criticized for its response times.

Overall, it's difficult to say which company is the most pro-consumer without more information about their specific policies and practices. However, based on industry trends and consumer feedback, it's clear that all of these companies have room for improvement in terms of privacy, pricing, and customer service.

| Instruction | I have competencies in remote sensing, machine learning, and water resource knowledge, what are the possible jobs I can occupy? What are the possible projects I can do? What companies I can work at? |
|---|---|
| Preference | BBA (Knowledgeable, Informative, Friendly) |

**Preference Prompting:** Hey there, fellow PhD student in remote sensing, machine learning, and water resource knowledge!

With your skills in these areas, the possibilities are endless for the types of jobs you could land and the projects you could undertake. You could work for a government agency, a private company, or even start your own consulting business.

As for companies you could work at, there are many options out there. For example, you could work for a company that specializes in water resource management, such as a company that provides water treatment solutions or a company that develops software for water resource management.

Or, you could work for a company that uses remote sensing technology to monitor natural resources, such as a company that provides satellite imagery for agriculture or a company that develops software for forestry management.

And let's not forget about the possibilities for research projects! You could work on a project that uses machine learning to improve the accuracy of remote sensing data, or a project that uses water resource knowledge to develop new water treatment technologies.

So, the sky's the limit for you, my friend! Keep exploring and learning, and who knows what amazing things you'll be able to accomplish.

Cheers!

**Personalized Soup:** Hey there, fellow PhD student in remote sensing, machine learning, and water resource knowledge!

With your skills in these areas, the possibilities are endless for the types of jobs you could land and the projects you could undertake. You could work for a government agency, such as the National Aeronautics and Space Administration (NASA) or the National Oceanic and Atmospheric Administration (NOAA), where you could use your expertise in remote sensing to monitor and analyze environmental changes.

Alternatively, you could work for a private company, such as a consulting firm or a tech company, where you could use your skills in machine learning to develop new algorithms and tools for data analysis.

As for projects, you could work on a research project that combines your expertise in remote sensing, machine learning, and water resource knowledge. For example, you could develop a machine learning model that predicts water availability in a specific region based on remote sensing data.

As for companies you could work at, there are many options! Some companies that might be a good fit for your skills include Google, Amazon, and Microsoft, among others.

So, what do you say? Ready to take on the world with your PhD in remote sensing, machine learning, and water resource knowledge? Let's do this!

**MPD:** Hey there, fellow PhD student in remote sensing, machine learning, and water resource knowledge! I'm so glad you're here, because I have some exciting information to share with you.

First off, let's talk about the possible jobs you could occupy with your skills. You could work as a research scientist, where you can use your expertise in remote sensing and machine learning to analyze data and develop new models. You could also work as a water resource specialist, where you can use your knowledge of water resources to help manage and protect water resources.

As for possible projects, you could work on developing new algorithms for remote sensing data analysis, or you could work on developing new models for water resource management. You could also work on developing new applications of machine learning in water resource management.

As for companies you could work at, there are many options! You could work at a research institution, a government agency, a consulting firm, or a private company.

Now, let's talk about the fun side of things. You could have a blast working on a project that involves analyzing satellite images of cute animals, like pandas or koalas. Or you could have a blast working on a project that involves developing a machine learning model to predict the best time to go swimming in a lake based on weather conditions.

So, there you have it, my friend! I hope this information was helpful and informative. Let's grab a cup of coffee and chat more about your research and your future career.

| Instruction | Antennas that transmit an equal amount of energy in the horizontal direction are called |
|---|---|
| Preference | BBB (Knowledgeable, Informative, Unfriendly) |

**Preference Prompting:** In the context of electromagnetic wave propagation, antennas that transmit an equal amount of energy in the horizontal direction are known as isotropic antennas. These antennas are idealized as having a radiation pattern that is uniform in all directions, and they are often used as a reference point in the analysis of more complex antenna systems.

The concept of isotropy is important in the study of electromagnetic waves because it allows us to make certain assumptions about the behavior of these waves in different directions. For example, if we know that a particular antenna has a certain radiation pattern in the horizontal direction, we can use this information to predict its behavior in the vertical direction.

In practice, it is difficult to create an antenna that is truly isotropic, as the physical properties of materials and the design of the antenna itself can introduce directional dependencies in the radiation pattern. However, by carefully controlling the shape and size of the antenna, it is possible to come close to achieving isotropy, and this can be useful in the analysis of more complex antenna systems.

In summary, isotropic antennas are important in the study of electromagnetic wave propagation because they allow us to make certain assumptions about the behavior of these waves in different directions. While it is difficult to create an antenna that is truly isotropic, it is possible to come close to achieving this goal, and this can be useful in the analysis of more complex antenna systems.

**Personalized Soup:** In order to transmit an equal amount of energy in the horizontal direction, the antennas must be properly designed and aligned. This is known as isotropic radiation, and it is achieved by using dipole antennas, which are symmetrical in shape and radiate energy equally in all directions. The use of isotropic radiation is important in the field of radio propagation, as it allows for the accurate measurement of signal strength and the calculation of signal attenuation due to obstructions such as buildings and terrain features.

**MPD:** As a PhD student in the field of electromagnetic wave propagation, I can confirm that the term "isotropic" is used to describe antennas that transmit an equal amount of energy in all directions. These antennas are often used in theoretical analysis and simulations to model the behavior of real-world antennas in a simplified manner.

In the context of antenna design, isotropy is an important property to consider, as it allows for the efficient use of energy and the minimization of interference between different signals. In practice, isotropic antennas are often used as a reference point in the analysis of antenna performance, as they provide a baseline for comparison with more complex, real-world antennas.

However, it is important to note that isotropic antennas are a theoretical construct and do not exist in reality. In practice, real-world antennas have certain properties that make them more suitable for specific applications, such as directional antennas that are designed to transmit energy in a specific direction, or omnidirectional antennas that are designed to transmit energy in all directions.

Overall, the concept of isotropy is an important one in the field of electromagnetic wave propagation, and a deep understanding of this concept is essential for the design and analysis of antennas and other electromagnetic systems.

| Instruction | How do you treat plantar fasciitis? |
|---|---|
| Preference | BBB (Knowledgeable, Informative, Unfriendly) |

**Preference Prompting:** Plantar fasciitis is a common condition that affects the heel and foot. It is caused by inflammation of the plantar fascia, a thick band of tissue that runs from the heel bone to the toes. Treatment for plantar fasciitis typically involves a combination of self-care measures and medical interventions.

Self-care measures include rest, ice, stretching exercises, and physical therapy. It is important to avoid activities that cause pain and to wear supportive shoes. Over-the-counter pain relievers, such as ibuprofen or naproxen, may also be helpful in reducing pain and inflammation.

In addition to self-care measures, medical interventions may be necessary to treat plantar fasciitis. These may include physical therapy, stretching exercises, and orthotics. In severe cases, corticosteroid injections or surgery may be considered.

It is important to work with a healthcare provider to develop a treatment plan that is appropriate for your individual needs. A healthcare provider can also help to determine the underlying cause of your plantar fasciitis and provide appropriate treatment.

It is important to note that plantar fasciitis can be a chronic condition, and it may take time to find the most effective treatment plan. However, with proper treatment and self-care, most people with plantar fasciitis can find relief from their symptoms and return to their normal activities.

**Personalized Soup:** Plantar fasciitis is a common condition that affects the heel and foot. It is caused by inflammation of the plantar fascia, a thick band of tissue that runs from the heel bone to the toes. Treatment for plantar fasciitis typically involves a combination of self-care measures and medical interventions.

Self-care measures include rest, ice, stretching, and physical therapy. Ice can help reduce inflammation and relieve pain, while stretching can help loosen the plantar fascia and improve flexibility. Physical therapy can also be helpful in improving foot mechanics and strengthening the muscles that support the foot.

In addition to self-care measures, medical interventions may be necessary to relieve symptoms and promote healing. These may include over-the-counter pain relievers, such as ibuprofen or naproxen, to reduce pain and inflammation. Injecting corticosteroids into the affected area can also provide relief from symptoms.

In severe cases, surgery may be necessary to relieve symptoms. However, this is typically a last resort and is only considered if other treatments have failed.

It is important to note that plantar fasciitis can be a chronic condition, and it may take several months of treatment to achieve relief from symptoms. It is important to follow the advice of your healthcare provider and to be patient and persistent in your efforts to treat the condition.

**MPD:** Plantar fasciitis is a common condition that affects the heel and foot. It is caused by inflammation of the plantar fascia, a thick band of tissue that runs from the heel bone to the toes. Treatment for plantar fasciitis typically involves a combination of self-care measures and medical interventions.

Self-care measures include rest, ice, stretching exercises, and physical therapy. These measures can help reduce inflammation and relieve pain. In addition, wearing supportive shoes with good arch support and cushioning can also help alleviate symptoms. Medical interventions for plantar fasciitis may include non-steroidal anti-inflammatory drugs (NSAIDs), such as ibuprofen or naproxen, to reduce inflammation and pain. Corticosteroid injections can also be used to reduce inflammation and relieve pain. In severe cases, surgery may be considered, but this is typically a last resort.

It is important to note that treatment for plantar fasciitis can take time, and it is important to follow a consistent treatment plan to achieve the best results. If you are experiencing symptoms of plantar fasciitis, it is important to see a healthcare provider for a proper diagnosis and treatment plan.

Table 8: Example generations for each of the eight distinct preferences from MPD.

| Instruction | Why is it a bad idea to give a mouse a cookie? |
| --- | --- |
| AAA (Elementary, Concise, Friendly) | Giving a mouse a cookie might seem like a good idea, but it's actually not a great idea. You see, if you give a mouse a cookie, it might want to ask for a glass of milk. And if you give the mouse a glass of milk, it might want to ask for a spoon. And if you give the mouse a spoon, it might want to start eating with it. And if it starts eating with a spoon, it might make a big mess. So, it's best to just give the mouse a cookie and not give it any more food. |
| ABA (Elementary, Informative, Friendly) | Giving a mouse a cookie can be a bad idea because mice might get addicted to the taste of the cookie and keep coming back for more, which can cause damage to your house. |
| ABA (Elementary, Concise, Friendly) | Giving a mouse a cookie might seem like a good idea, but it's actually not a very smart thing to do. You see, mice are small creatures that can easily get lost in a big, scary world. If you give a mouse a cookie, it might not know how to find its way back home. Plus, if the mouse eats the cookie, it might not have enough energy to explore and find its way back home. It's important to be careful and make sure that everyone, even small creatures like mice, are safe and happy. |
| ABB (Elementary, Informative, Unfriendly) | It's a bad idea to give a mouse a cookie because mice are small and can easily get stuck in the cookie. They might also eat too many cookies and become sick. Additionally, if there are many mice in the house, they could chew through wires and cause problems with the electricity. |
| BAA (Knowledgeable, Concise, Friendly) | Giving a mouse a cookie could lead to a domino effect of events that could potentially harm the mouse, such as weight gain, diabetes, and even death. It's important to remember that animals, including mice, have specific dietary needs to maintain their health and well-being. So, it's best to stick to giving them appropriate foods and avoid giving them human foods like cookies. |
| BAB (Knowledgeable, Concise, Unfriendly) | Giving a mouse a cookie can lead to unintended consequences, such as cookie crumbs attracting pests and compromising the integrity of the experiment. |
| BBA (Knowledgeable, Informative, Friendly) | Giving a mouse a cookie may seem like a harmless act, but it can actually lead to some unexpected consequences. Firstly, mice are known to be curious creatures, and once they discover a cookie, they may become obsessed with it, leading to overeating and potential health problems. Additionally, mice are known to be prolific breeders, and if they have access to a steady supply of cookies, they may reproduce at an accelerated rate, leading to an explosion in their population. This can cause damage to your home, as mice are known to chew through walls and electrical wires, potentially causing fires and other hazards. Finally, mice are also known to be attracted to certain types of food, such as chocolate, which can be toxic to them, leading to serious health problems. So, while it may seem like a fun idea to give a mouse a cookie, it's important to consider the potential consequences before doing so. |
| BBB (Knowledgeable, Informative, Unfriendly) | It is a bad idea to give a mouse a cookie because it can lead to unintended consequences and may not be in the best interest of the mouse. For example, if the mouse eats the cookie, it may experience digestive issues due to the high sugar content. Additionally, the cookie may contain ingredients that are harmful to the mouse, such as artificial colors or preservatives.

Furthermore, giving a mouse a cookie may not be ethical or humane. Mice are small animals that have specific dietary needs, and providing them with a food that is not appropriate for their species can lead to health problems and even death.

In summary, it is important to consider the potential risks and ethical implications before giving a mouse a cookie, and to provide them with a diet that is appropriate for their species. |

