# OpenReview forum: "On Orchestrating Personalized LLMs"
_ICLR.cc/2025/Conference — Submitted to ICLR 2025_

### Official Review · Reviewer_6WP3 · 2024-10-28

**Soundness:** 3
**Presentation:** 4
**Contribution:** 3
**Rating:** 6
**Confidence:** 4

**Summary:**

The authors propose a method for merging LLM outputs on a per-token level to improve LLM personalization, using an additional control model to control how to combine multiple expert model outputs. The control model is trained via an online RL algorithm (the authors show results with PPO and REBEL). The proposed method outperforms other personalization methods (prompting, souping) based on both llm-as-a-judge and human experiments. Additional ablations justify some design decisions such as merging in the probability space and transforming the reward via a bradley-terry model.

**Strengths:**

- The core idea is straightforward (merging output probabilities and weighting based on preferences) and the paper does a good job of exploring various design decisions around it (how to train the control model, logits vs probabilities, RL algorithm choice), and is clearly laid-out and written.
- The performance improvements seem robust, with llm-as-a-judge and human evaluation results both suggesting the proposed approach improves over prior work.
- The discussion about inference efficiency is interesting and raises a good point about parameter merging-based approaches, although I think the fact that this method requires a potentially higher memory footprint for inference is a drawback.

**Weaknesses:**

- The method has some limitations addressed in the paper: having to train on all preference pairs (which may make scaling to a large number of preferences less feasible), having to retrain the model if new preference types are introduced, and having a larger memory footprint compared to parameter merging methods at inference time.
- Results shown are always winrate over all prompts, which makes it details on where and how the method is improving unclear beyond a handful of qualitative examples. I would be curious if breaking down results per-preference dimension (shown in Table 1) and comparing with baselines would yield any interesting observations - perhaps some methods are better than others at particular preference dimensions?
- Some clarification would be useful around the methods - see my questions below.

Overall, I think the approach proposed is solid, the results reasonable, and so lean accept for this paper.

**Questions:**

- Is preference prompting used for the MPD (Uniform) setting (for generating responses)? I don’t quite understand how this could yield better personalization otherwise.
- How many expert models are trained? Is it one per row in Table 1, or one per A/B pair? I found this a bit unclear.
- I am curious about how well an expert-only baseline would do, where we compare just using one of the expert models for generation - do we gain anything by merging the output distributions token-by-token, or does this perform similarly to picking the best output from relevant expert models?
- It would be interesting to see some analysis on the weights that get chosen by the control model.

---

> ### Author Response · Authors · 2024-11-22
> **Response to Reviewer 6WP3 (Part 1)**
>
> Thank you for your encouraging comments and constructive feedback. We address individual points below.
>
> > The method has some limitations addressed in the paper: having to train on all preference pairs (which may make scaling to a large number of preferences less feasible), having to retrain the model if new preference types are introduced, and having a larger memory footprint compared to parameter merging methods at inference time.
>
> Thank you for the comment. We would like to provide some thoughts on the limitations.
>
> **Train on all preference pairs**: Note that only a single PCM is trained with RL where we uniformly randomly sample preference combinations during training. This can be considered as an RL training procedure with diverse initial state distributions (i.e. preference combinations in addition to instructions are provided to the PCM as prompts). The training ensures that one RL procedure can learn to perform well on average across all preference combinations. We note that we do not need to perform training or maintain a separate PCM per preference combination which is indeed not tractable.
>
> **Large memory footprint**: We agree with the reviewer that the **memory** footprint of MPD is larger than weight merging baselines but the **inference time** is actually more efficient as discussed between Line 489-506. One potential memory footprint reduction technique is to use lower precision for the expert models while still keeping full precision of PCM. Because the output probability is weighted averaged / ensembled, the output responses from MPD should be more robust to lower precision.
>
> > Results shown are always winrate over all prompts, which makes it details on where and how the method is improving unclear beyond a handful of qualitative examples. I would be curious if breaking down results per-preference dimension (shown in Table 1) and comparing with baselines would yield any interesting observations - perhaps some methods are better than others at particular preference dimensions?
>
> Thank you for the great suggestion and below we provide the win rate statistics of MPD against two baselines on the individual preference dimensions. It can be seen that MPD indeed performs differently: 1A (elementary) and 2A (concise) consistently outperforming baselines, whereas 1B (knowledgeable) being slightly worse than baselines.
>
> |Preference Dimension|vs Preference Prompting|vs Personalized Soup|
> |-|-|-|
> |1A|53.5%|53.5%|
> |1B|47.5%|45.8%|
> |2A|62.3%|59.3%|
> |2B|51.0%|48.8%|
> |3A|51.0%|49.5%|
> |3B|51.3%|49.5%|
>
> > Is preference prompting used for the MPD (Uniform) setting (for generating responses)? I don’t quite understand how this could yield better personalization otherwise.
>
> Yes, the reviewer is correct. We used the same preference prompting for Personalized Soup and MPD to ensure a fair comparison. Intuitively, it is possible that MPD (Uniform) outperforms Personalized Soup. The difference is that Personalized Soup averages model **parameters** and MPD averages model **outputs**. The results show that averaging model outputs performs better and it is not very surprising since averaging outputs might be more interpretable than averaging parameters.
>
> > How many expert models are trained? Is it one per row in Table 1, or one per A/B pair? I found this a bit unclear.
>
> Sorry for the confusion. There are six expert models trained (one per row in Table 1). We will explicitly mention this in our revision.
>
> > I am curious about how well an expert-only baseline would do, where we compare just using one of the expert models for generation - do we gain anything by merging the output distributions token-by-token, or does this perform similarly to picking the best output from relevant expert models?
>
> Thank you for the suggestion. In our preliminary experiments, we found that when our actual reward is only based on one of the preference dimensions, the PCM would learn to allocate a weight of 1 to that dimension and 0 to the other dimensions after training. This suggests that the expert is indeed very strong in personalization for the individual preference dimension and our RL-based approach will learn to converge to the correct expert.

---

> > ### Author Response · Authors · 2024-11-22
> > **Response to Reviewer 6WP3 (Part 2)**
> >
> > > It would be interesting to see some analysis on the weights that get chosen by the control model.
> >
> > Thank you for the question and below we provide two observations on the weights the control model outputs.
> >
> > First, at some token decoding step, the distribution of weights is less important and can be arbitrary. For example, when a long word that consists of multiple tokens already has its first token decoded, subsequent token output probability distribution from individual preference experts almost entirely agree with each other until finishing decoding of that word, which makes weight distribution arbitrary.
> >
> > Second, we find that compared to MPD (Uniform), MPD achieves higher win rate on friendly / humorous dimension. The average weight for those generations on the friendly / humorous dimension accounts for more than 60% of the total weight whereas the other two dimensions each contribute to about 20%. This suggests that on an aggregated scale, allocating more weights to a preference dimension could lead to better performance on that dimension.
> >
> > [1] Rame, Alexandre, Guillaume Couairon, Corentin Dancette, Jean-Baptiste Gaya, Mustafa Shukor, Laure Soulier, and Matthieu Cord. "Rewarded soups: towards pareto-optimal alignment by interpolating weights fine-tuned on diverse rewards." Advances in Neural Information Processing Systems 36 (2024).

---

> > > ### Comment · Reviewer_6WP3 · 2024-11-24
> > >
> > > Hi, thank you for the detailed response! I've read it carefully along with the other reviews and am keeping my score.

---

> > > > ### Author Response · Authors · 2024-11-24
> > > > **Response to Reviewer 6WP3**
> > > >
> > > > Dear Reviewer 6WP3,
> > > >
> > > > Thank you for your time and feedback! If there are any other aspects we can help clarify please let us know.
> > > >
> > > > Regards,
> > > >
> > > > Authors

---

### Official Review · Reviewer_DVt6 · 2024-10-31

**Soundness:** 3
**Presentation:** 3
**Contribution:** 3
**Rating:** 6
**Confidence:** 4

**Summary:**

This paper presents Merged Preference Dimensions (MPD), a method for aligning large language models (LLMs) with personalized human preferences. Instead of fine-tuning or merging model parameters, MPD dynamically merges the outputs of specialized expert LLMs at the token level.

A lightweight Preference Control Model (PCM) translates the preference description and current context into weights for combining the next-token predictions of the experts. This approach offers a black-box, parallelizable solution for personalized text generation. Empirical results on Koala and UltraFeedback datasets show that MPD achieves higher pairwise win rates compared to existing preference merging and prompting techniques.

The idea is novel and interesting, with the PCM acting as a orchestra conductor, while the models are like individual players.

**Strengths:**

* **Novelty:**  MPD introduces the innovative idea of context-dependent, token-level merging of expert LLM outputs, which differentiates it from previous parameter merging techniques.
* **Practicality:** The black-box nature of MPD and its applicability to different architectures make it more practical for real-world scenarios.  The parallelizability and lack of retraining for preference changes are further advantages.
* **Efficiency:** The demonstrated inference speed advantage due to parallelization is a valuable contribution, particularly in contexts where multiple preferences need to be served concurrently.

**Weaknesses:**

* **Limited Preference Dimensions and Generalizability:** The selected preference dimensions appear arbitrary and lack sufficient justification. The paper does not adequately address the generalizability of MPD to a more diverse and nuanced range of human preferences. Given the limited number of dimensions explored, it remains unclear how MPD would perform with a substantially larger set, which would be necessary to capture the true complexity of human preferences.

* **Scalability of Preferences and Combinatorial Explosion:** The necessity of enumerating all preference combinations during training introduces a scalability bottleneck. While the paper acknowledges this limitation and briefly touches upon handling new preferences, the proposed approach does not convincingly demonstrate MPD's scalability to a higher-dimensional preference space. The combinatorial explosion inherent in an increasing number of dimensions raises concerns about the feasibility of training MPD with a realistically large preference set.

* **Justification for Tulu-7B and Model Size:** The rationale for selecting Tulu-7B as the base model is insufficiently explained. An evaluation of MPD using more widely adopted, powerful LLMs (e.g., Llama 3, Qwen2.5, Anthropic, GPT, etc,) would significantly strengthen the paper's conclusions. Furthermore, the relatively small size of the tested model (7B) raises questions about the robustness and generalizability of the learned merging strategy. The observed win rates near 50% suggest a limited effect size, potentially attributable to the PCM's capacity.

* **Insufficient Detail on Reward Models and Training:** The paper lacks crucial details about the reward models, including their architecture, training data, and evaluation metrics. This lack of transparency hinders reproducibility and a thorough understanding of the evaluation process. Similarly, essential details about the LoRA training procedure, such as rank, alpha value, batch size, and optimization parameters, are omitted. These hyperparameters can significantly influence the performance and output distribution of LLMs, making their disclosure essential for reproducibility and analysis.

**Questions:**

* The paper relies on LLM-as-a-judge [2] for evaluation, a methodology known to be susceptible to positionality bias. Could you please talk about whether measures were taken to mitigate this bias, such as comparing both A vs. B and B vs. A, as recommended in prior work such as [1]. This omission raises concerns about the reliability of the reported win rates, (though human evaluation has been conducted as well).
* How does MPD's performance scale with an increasing number of preference dimensions? What strategies are being considered to mitigate the combinatorial explosion of preference combinations during training?
* In comparison to methods that employ smaller LLMs to generate natural language rules for controlling larger models (e.g., [3]), what are the specific advantages of MPD's token-level probability merging approach? Why is this preferable to using freeform natural language rules generated by a smaller LLM to guide a single, more powerful LLM?
* What was the rationale for choosing Tulu-7B as the base model? Have experiments been conducted with other similarly sized LLMs (e.g., Llama 3 8B, Qwen2.5 7B) or larger, more powerful models (Gemini, Llama 70B, Sonnet, etc)? If so, how do the results compare?
* Please provide comprehensive details on the reward models, specifying their architecture, training data, and evaluation metrics. How was the accuracy and calibration of these models ensured?
* Please provide detailed information about the LoRA training procedure for both the expert models and the PCM, including the rank, alpha value, batch size, and other relevant hyperparameters.


[1] https://arxiv.org/abs/2305.14314

[2] https://arxiv.org/abs/2306.05685

[3] https://arxiv.org/abs/2410.03731

---

> ### Author Response · Authors · 2024-11-22
> **Response to Reviewer DVt6 (Part 1)**
>
> Thank you for the detailed review and constructive feedback! We address individual points below.
>
> > Limited Preference Dimensions and Generalizability: The selected preference dimensions appear arbitrary and lack sufficient justification. The paper does not adequately address the generalizability of MPD to a more diverse and nuanced range of human preferences. Given the limited number of dimensions explored, it remains unclear how MPD would perform with a substantially larger set, which would be necessary to capture the true complexity of human preferences.
>
> We first highlight that there is no standard benchmark or preference dimensions commonly used in related work [1, 2, 3]. For a fair comparison, we closely follow the experimental setup of [1] which is one of the first works using preference experts for multi-objective personalization. As discussed in [1], the three preference dimensions correspond to (expertise, informativeness, style) with two opposite options for each dimension (A and B as seen in Table 1). We also note that other related work such as [3] also only experiments with three preference dimensions. We chose to follow the setting in [1] for a fair comparison to their approach.
>
> > Scalability of Preferences and Combinatorial Explosion: The necessity of enumerating all preference combinations during training introduces a scalability bottleneck. While the paper acknowledges this limitation and briefly touches upon handling new preferences, the proposed approach does not convincingly demonstrate MPD's scalability to a higher-dimensional preference space. The combinatorial explosion inherent in an increasing number of dimensions raises concerns about the feasibility of training MPD with a realistically large preference set.
>
> > How does MPD's performance scale with an increasing number of preference dimensions? What strategies are being considered to mitigate the combinatorial explosion of preference combinations during training?
>
> Thank you for the great question and we fully acknowledge that MPD is trained on all preference combinations. We would like to highlight that, however, only a single PCM is trained with RL where we uniformly randomly sample preference combinations during training. This can be considered as an RL training procedure with diverse initial state distributions (i.e. preference combinations in addition to instructions are provided to the PCM as prompts). The training ensures that one RL procedure can learn to perform well on average across all preference combinations. We note that we do not need to perform training or maintain a separate PCM per preference combination which is indeed not tractable. In contrast, Personalized Soup indeed needs to perform model merging at inference time for all possible combinations, which can be intractable.
>
> Besides, as discussed briefly on Line 511, other multi-objective approaches such as [2] that require training are also trained on all preference combinations. In fact, the training of MPD is more efficient than other approaches since it does not calculate gradients for the multiple expert models. The gradient is only calculated on the much smaller PCM model.
>
> > Justification for Tulu-7B and Model Size: The rationale for selecting Tulu-7B as the base model is insufficiently explained. An evaluation of MPD using more widely adopted, powerful LLMs (e.g., Llama 3, Qwen2.5, Anthropic, GPT, etc,) would significantly strengthen the paper's conclusions. Furthermore, the relatively small size of the tested model (7B) raises questions about the robustness and generalizability of the learned merging strategy. The observed win rates near 50% suggest a limited effect size, potentially attributable to the PCM's capacity.
>
> > What was the rationale for choosing Tulu-7B as the base model? Have experiments been conducted with other similarly sized LLMs (e.g., Llama 3 8B, Qwen2.5 7B) or larger, more powerful models (Gemini, Llama 70B, Sonnet, etc)? If so, how do the results compare?
>
> We used Tulu-7B because we want to have a direct and fair comparison with [1] from which we also adopted the experimental setup. To ensure PCM is not limiting the performance, we also carry out an experiment where we directly finetune the base Tulu-7B model with the same reward function and RL learning algorithm on all preference combinations. This should be considered as an upper bound of MPD. MPD has a 49.15% win rate against this approach, suggesting that MPD is achieving strong personalization results.

---

> > ### Author Response · Authors · 2024-11-22
> > **Response to Reviewer DVt6 (Part 2)**
> >
> > > Insufficient Detail on Reward Models and Training: The paper lacks crucial details about the reward models, including their architecture, training data, and evaluation metrics. This lack of transparency hinders reproducibility and a thorough understanding of the evaluation process. Similarly, essential details about the LoRA training procedure, such as rank, alpha value, batch size, and optimization parameters, are omitted. These hyperparameters can significantly influence the performance and output distribution of LLMs, making their disclosure essential for reproducibility and analysis.
> >
> > > Please provide comprehensive details on the reward models, specifying their architecture, training data, and evaluation metrics. How was the accuracy and calibration of these models ensured?
> >
> > > Please provide detailed information about the LoRA training procedure for both the expert models and the PCM, including the rank, alpha value, batch size, and other relevant hyperparameters.
> >
> > Sorry for missing these implementation details. We mentioned briefly on Line 313 that our reward models and expert models were trained in the same way as [1]. We also used their codebase (https://github.com/joeljang/RLPHF). Specifically, the models have the same architecture of Tulu-7B and are LoRA finetuned. The LoRA rank is 16 with alpha being 32. The LoRA adapters have a 5% chance of dropout with no bias terms. The training batch size is 16 with a learning rate of 1.5e-5 using AdamW optimizer implemented in Pytorch.
> >
> > The training data for the reward models are pairwise responses to 10k GPT4-Alpaca prompts labeled by GPT4 deciding which response is better. The reward model was trained for one epoch on the dataset and binary classification accuracy was evaluated on all preference dimensions with average accuracy being around 75%.
> >
> > For PCM, as discussed between Line 316-318, it is a 160M Llama-based model (https://huggingface.co/JackFram/llama-160m). The small size is attributed to a small hidden size and fewer number of layers compared to Llama 2. The last linear layer of this model is replaced with a much smaller linear layer of size number of preference dimensions ($n = 3$). The LoRA and training configuration is the same as reward and expert models.
> >
> > Thank you for the valuable suggestion and we will incorporate the above details in the final version of the paper.
> >
> > > The paper relies on LLM-as-a-judge [2] for evaluation, a methodology known to be susceptible to positionality bias. Could you please talk about whether measures were taken to mitigate this bias, such as comparing both A vs. B and B vs. A, as recommended in prior work such as [1]. This omission raises concerns about the reliability of the reported win rates, (though human evaluation has been conducted as well).
> >
> > Thank you for the comment. Yes, as discussed in Appendix A.2, for **both GPT4 and human evaluation**, we randomly flip the order of two responses to reduce the positional bias. We will explicitly discuss this in the main text in our revision.
> >
> > > In comparison to methods that employ smaller LLMs to generate natural language rules for controlling larger models (e.g., [3]), what are the specific advantages of MPD's token-level probability merging approach? Why is this preferable to using freeform natural language rules generated by a smaller LLM to guide a single, more powerful LLM?
> >
> > Thank you so much for the thought-provoking question! The approach [4] proposes is a very promising direction for achieving personalization. We will definitely include this in our related work section. We were not aware of this work since it appeared *after* our submission deadline.
> >
> > We believe this approach is compatible with and complementary to the token-level probability merging approach. This is because natural language rules and instructions can also be broken down and provided to expert models of MPD and we leave the exploration of this combination in future work.
> >
> > Additionally, we hypothesize the MPD approach may be more beneficial than free form natural language rules when the base model is less capable. Intuitively, being able to understand and leverage the natural language rules require strong instruction-following ability of the base model. Generating natural language rules also requires the “small” model to be competitive. In the experiment section of [4], it can be seen that the base models $M_L$ are either 70B Llama or frontier closed-source models. The model that is used to generate natural language rules is also a competitive Llama 3 8B model. For MPD, however, we demonstrate that we can control 7B base models with a very small 160M model. These results suggest that MPD is also a useful guiding approach.

---

> > > ### Author Response · Authors · 2024-11-22
> > > **Response to Reviewer DVt6 (Part 3)**
> > >
> > > [1] Jang, Joel, Seungone Kim, Bill Yuchen Lin, Yizhong Wang, Jack Hessel, Luke Zettlemoyer, Hannaneh Hajishirzi, Yejin Choi, and Prithviraj Ammanabrolu. "Personalized soups: Personalized large language model alignment via post-hoc parameter merging." arXiv preprint arXiv:2310.11564 (2023).
> > >
> > > [2] Rame, Alexandre, Guillaume Couairon, Corentin Dancette, Jean-Baptiste Gaya, Mustafa Shukor, Laure Soulier, and Matthieu Cord. "Rewarded soups: towards pareto-optimal alignment by interpolating weights fine-tuned on diverse rewards." Advances in Neural Information Processing Systems 36 (2024).
> > >
> > > [3] Shi, Ruizhe, Yifang Chen, Yushi Hu, ALisa Liu, Hannaneh Hajishirzi, Noah A. Smith, and Simon S. Du. "Decoding-time language model alignment with multiple objectives." arXiv preprint arXiv:2406.18853 (2024).
> > >
> > > [4] Shashidhar, Sumuk, Abhinav Chinta, Vaibhav Sahai, and Dilek Hakkani-Tür. "Unsupervised Human Preference Learning." arXiv preprint arXiv:2410.03731 (2024).

---

> > > ### Comment · Reviewer_DVt6 · 2024-11-24
> > > **Response to authors**
> > >
> > > Thank you for your clarifications and the reference implementation, which I believe is beneficial for reproducibility. I have increased my score as a reflection.

---

> > > > ### Author Response · Authors · 2024-11-24
> > > > **Response to Reviewer DVt6**
> > > >
> > > > Dear Reviewer DVt6,
> > > >
> > > > Thank you so much for increasing your recommendation. If there are any other aspects we can help clarify please let us know.
> > > >
> > > > Regards,
> > > >
> > > > Authors

---

### Official Review · Reviewer_XXCM · 2024-11-07

**Soundness:** 1
**Presentation:** 3
**Contribution:** 1
**Rating:** 5
**Confidence:** 5

**Summary:**

This paper introduces a novel framework, Merged Preference Dimensions (MPD), for dynamically aligning large language models (LLMs) with individual user preferences without re-training the entire model. By utilizing a lightweight Preference Control Model (PCM), it merges outputs from expert LLMs trained in specific preference dimensions (like humor or conciseness) at the token level. This approach allows for real-time customization of LLM outputs to suit individual preferences, demonstrating superior performance in adapting to user-specified dimensions compared to traditional methods. The framework is efficient, scalable, and requires no access to original training data, offering a flexible solution for personalized language generation.

**Strengths:**

1. The paper is well-written.
2. The problem of personalized LLMs is timely and interesting.

**Weaknesses:**

1. Significance of the Problem: The manuscript describes a solution for a "black-box method" using trained expert models for each preference dimension. However, the necessity and practicality of these black-box expert models are questionable as they do not seem to exist in practice. As described in the manuscript, the expert models are not pre-existing and are instead trained by the authors following the procedure from Jang et al. (2023). This approach contradicts the concept of using already available black-box models and raises questions about the real-world applicability of the proposed method.

2. Lack of Technical Novelty: Based on the approach described, I do not find significant innovation in this paper, especially in comparison to existing literature cited as [1-3], particularly [3]. The content in Section 3.1 looks similar as in [1], and Section 3.2 seems to extend existing work on RLHF and REBEL without introducing substantial new insights or advancements. This overlap with prior works might limit the perceived contribution of the current study.

3. Experimental Setup: The experimental design, as mentioned, appears to be flawed (refer to Question 2).

[1] Personalized soups

[2] Rewarded soups

[3] Decoding-Time Language Model Alignment with Multiple Objectives

**Questions:**

1. Figure 1 Query: The user input is stated as “I want a helpful, concise, and funny response!”. It is not clear why the weights for the three dimensions are set as 0.4, 0.1, and 0.5 respectively. How can this distribution of weights be explained?

2. Source of Weights: Regarding the Koala and Ultrafeedback datasets and the preference dimensions listed in Table 1, I am curious about how the authors derived the different lists of weights for these dimensions. Is it possible to obtain the ground truth for these weights, or are the weights generated by the Preference Control Model (PCM) interpretable?

3. PCM Training Process: The Final Reward is obtained by an average weighted sum of different Bradley-Terry Rewards. Why does the PCM employ a non-uniform weighting in this context? An explanation of the choice of weights used and their impact on the model’s output would provide deeper insights into the PCM’s functionality.

4. Computational Cost and Scalability: The training process for the Merged Preference Dimensions (MPD) framework appears to require loading multiple pre-trained LLMs as well as the PCM simultaneously. I am concerned about the computational cost and scalability of this training process. Could the authors provide details regarding the computational resources required and discuss the potential limitations in terms of scalability?

---

> ### Author Response · Authors · 2024-11-22
> **Response to Reviewer XXCM (Part 1)**
>
> Thank you for your valuable and insightful comments. We address individual points below.
>
> > Significance of the Problem: The manuscript describes a solution for a "black-box method" using trained expert models for each preference dimension. However, the necessity and practicality of these black-box expert models are questionable as they do not seem to exist in practice. As described in the manuscript, the expert models are not pre-existing and are instead trained by the authors following the procedure from Jang et al. (2023). This approach contradicts the concept of using already available black-box models and raises questions about the real-world applicability of the proposed method.
>
> Thank you for the great question! As discussed in Appendix A.1 Line 706, we trained the expert models ourselves only because [1] did not release trained checkpoints for the expert models. We also believe that black-box methods are more advantageous since it implies that we will not compute the gradient of expert models which makes the approach much more feasible if the expert models are too large or too expensive to be finetuned.
>
> We also give our thought on if expert models naturally exist. We are under the impression that there do not exist many pre-trained expert models in academia. However, we do believe expert or personalized models exist in the industry and have real-world applicability. For example, Character AI has a diverse set of characters, each has some unique personality and is tailored towards the audience with different personalization goals.
>
> > Lack of Technical Novelty: Based on the approach described, I do not find significant innovation in this paper, especially in comparison to existing literature cited as [1-3], particularly [3]. The content in Section 3.1 looks similar as in [1], and Section 3.2 seems to extend existing work on RLHF and REBEL without introducing substantial new insights or advancements. This overlap with prior works might limit the perceived contribution of the current study.
>
> Thank you for the references! We do not believe Section 3.1 looks similar to [1] or [2] since both work merge model weights instead of model outputs. We would appreciate it if the reviewer could elaborate on which specific parts are similar.
>
> For Section 3.2, we actually intend to make MPD training simple without introducing any novel RL components (this is not a new RLHF paper). In fact, our experiments demonstrate that any online RL algorithm such as PPO and REBEL can be applied for MPD training. Our main contribution is a framework and proof-of-concept where a small PCM that controls how token-level output probability from experts are merged can achieve strong results for multi-objective personalization.
>
> The main distinction between MPD and [3] is that [3] directly merges the output from expert models without any training of the merging weights. In our manuscript, this is an ablation of MPD, namely MPD (Uniform) where we uniformly merge the outputs from the experts. In Table 3, the results demonstrate that by further training a small PCM model, we can improve the win rate by more than 6%, suggesting the effectiveness of our approach compared to [3].

---

> > ### Author Response · Authors · 2024-11-22
> > **Response to Reviewer XXCM (Part 2)**
> >
> > > Figure 1 Query: The user input is stated as “I want a helpful, concise, and funny response!”. It is not clear why the weights for the three dimensions are set as 0.4, 0.1, and 0.5 respectively. How can this distribution of weights be explained?
> >
> > > Source of Weights: Regarding the Koala and Ultrafeedback datasets and the preference dimensions listed in Table 1, I am curious about how the authors derived the different lists of weights for these dimensions. Is it possible to obtain the ground truth for these weights, or are the weights generated by the Preference Control Model (PCM) interpretable?
> >
> > So sorry for the confusion. We first clarify that the weights are not fixed for a given preference. Instead, the weight is **dynamically** generated by the PCM **at each token**.
> >
> > Figure 1 is a pictorial illustration of our method. Intuitively speaking, at the beginning of a response, setting up a funny tone could be important for the response to be funny. Therefore, the weight for the funny expert may be higher than other dimensions decided by the learned PCM. But we note that the weights can be updated in the later decoding steps.
> >
> > The weights are learned with online RL by optimizing the reward function defined in Equation (2) of the paper. Since there is no ground truth of free-form generations, we cannot obtain the ground truth for these weights.
> >
> > In terms of interpretability, we have two observations. First, at some token decoding step, the distribution of weights is less important and can be arbitrary. For example, when a long word that consists of multiple tokens already has its first token decoded, subsequent token output probability distribution from individual preference experts almost entirely agree with each other until finishing decoding of that word, which makes weight distribution arbitrary. Second, we find that compared to MPD (Uniform), MPD achieves higher win rate on friendly / humorous dimension. The average weight for those generations on the friendly / humorous dimension accounts for more than 60% of the total weight whereas the other two dimensions each contribute to about 20%. This suggests that on an aggregated scale, allocating more weights to a preference dimension could lead to better performance on that dimension.
> >
> > > PCM Training Process: The Final Reward is obtained by an average weighted sum of different Bradley-Terry Rewards. Why does the PCM employ a non-uniform weighting in this context? An explanation of the choice of weights used and their impact on the model’s output would provide deeper insights into the PCM’s functionality.
> >
> > Sorry for the confusion. The final reward is an **unweighted** average of different Bradley-Terry rewards. Additionally, PCM does not control or influence the reward calculation. We will make this clearer in our revision.
> >
> > > Computational Cost and Scalability: The training process for the Merged Preference Dimensions (MPD) framework appears to require loading multiple pre-trained LLMs as well as the PCM simultaneously. I am concerned about the computational cost and scalability of this training process. Could the authors provide details regarding the computational resources required and discuss the potential limitations in terms of scalability?
> >
> > Thank you for the great question and we are also mindful about the efficiency of our approach! Because MPD does not calculate gradients on the parameters of the large expert models, and only optimizes the parameters of a small PCM, the training of MPD is computationally scalable. That is, the expert models are only used for inference rather than training. This is the key to keep our approach efficient since for a very large expert model, gradient calculation and backpropagation could incur 2-3x more memory usage than that needed to store the model parameters alone.
> >
> > Additionally, on Line 489-506, we also discuss the inference efficiency for MPD compared to weight merging baselines such as Personalized Soup. We show that although MPD incurs more computational cost at inference, it can actually be more efficient since it is more suitable for parallelism. This parallelism helps both the training and inference because during training, we also generate responses and use online-RL methods to optimize the weights of PCM.

---

> > > ### Comment · Reviewer_XXCM · 2024-11-25
> > >
> > > Thanks for your response. I still have three remaining concerns regarding the response.
> > >
> > > > Significance of the Problem
> > >
> > > I reserve my concerns that the expert models are difficult to obtain. For example, Character AI has a diverse set of characters, each with a unique personality. However, in this paper, expert models are actually designed for different dimensions. Therefore, it seems that expert models should be customized rather than reused from existing ones.
> > >
> > > > Source of Weights
> > >
> > > I still don't understand why "The weights are learned with online RL by optimizing the reward function defined in Equation (2) of the paper."
> > >
> > > > PCM Training Process
> > >
> > > Do you mean that the final objective is unweighted average? However, during the training process, PCM gives weighted average reward. Why?

---

> > > > ### Author Response · Authors · 2024-11-25
> > > > **Response to Reviewer XXCM**
> > > >
> > > > Dear Reviewer XXCM,
> > > >
> > > > Thank you for reading our response and we are happy to clarify the concerns above.
> > > >
> > > > > I reserve my concerns that the expert models are difficult to obtain. For example, Character AI has a diverse set of characters, each with a unique personality. However, in this paper, expert models are actually designed for different dimensions. Therefore, it seems that expert models should be customized rather than reused from existing ones.
> > > >
> > > > Thank you for the great thought again. Please correct us if we misunderstood: it seems that the reviewer is concerned with the fact that different character models have specific personality and therefore it is not a “pure” expert that only specializes in a single preference dimension that we care about.
> > > >
> > > > For the above case, it is still possible to achieve desired personalization from these character models. As long as we have access to the *reward models for the preference dimensions* that we care about, given a set of character models, we can run the same RL algorithm in the paper to maximize the reward of responses for those preference dimensions.
> > > >
> > > > Very intuitively speaking, different preference dimensions together lead to a high dimensional preference space. The “pure” expert models form the most straightforward “basis vectors” for this space whereas many character models are not fully aligned with the preference dimensions but may still cover the space.
> > > >
> > > > Alternatively, one may also consider simulating expert models for different preference dimensions via elaborate prompting from a single base model. This also makes the expert models easy to obtain and maintain.
> > > >
> > > > > I still don't understand why "The weights are learned with online RL by optimizing the reward function defined in Equation (2) of the paper."
> > > >
> > > > > Do you mean that the final objective is unweighted average? However, during the training process, PCM gives weighted average reward. Why?
> > > >
> > > > Sorry for the confusion caused and please allow us to explain it better. To directly answer the question: The final reward objective is always an **unweighted average** of **rewards** from different preference dimensions. PCM does not control or give weights to how the average reward is calculated. Instead, PCM gives weights on how to average the **next token probability distribution** of the expert models at every token decoding step.
> > > >
> > > > We realize that Figure 2 and the text associated with it can be slightly confusing. We will make it clear what PCM output weights are for. Below we also provide a more detailed explanation of how inference and training work for MPD.
> > > >
> > > > **Inference**: At every decoding step, we first obtain the next token probability distribution from the expert models. Then, PCM outputs a set of weights that sum to 1. A new next token probability distribution is obtained by a weighted average (Equation 1). The process is repeated until an EOS token is generated. Note that PCM may output different sets of weights at different decoding steps.
> > > >
> > > > **Training**: We first run inference with the expert models and current version of PCM using the inference procedure above. This gives us a response $y$. We also have a reference response $y_{\text{ref}}$, and reward models for each preference dimension. Then the reward of $y$ for each preference dimension is calculated with the Bradley-Terry model (Equation 2). The final reward of $y$ is a simple average of the rewards from each preference dimension (Line 245). To update the PCM, we use REBEL and follow the equation on Line 252-254.
> > > >
> > > > Please let us know if any issues remain and/or if there are any additional clarifications we can provide!
> > > >
> > > > Regards,
> > > >
> > > > Authors

---

> > > > > ### Comment · Reviewer_XXCM · 2024-11-26
> > > > >
> > > > > Thanks for your response.
> > > > >
> > > > > For the first question, I still hold my concern that black-box expert models are seldom available.
> > > > >
> > > > > For the second question, I understand that PCM can help better align with the final preference in the decoding process, correct?
> > > > >
> > > > > Anyway, I appreciate the authors' efforts and raise my score.

---

> > > > > > ### Author Response · Authors · 2024-11-27
> > > > > > **Response to Reviewer XXCM**
> > > > > >
> > > > > > Dear Reviewer XXCM,
> > > > > >
> > > > > > Thank you for increasing your score and we appreciate the valuable discussion with you on the first question. For the second question, the reviewer is correct that PCM indeed is designed to better align with and optimize the final multi-objective preference via orchestrating the individual expert models. Please let us know if there is anything we can clarify further.
> > > > > >
> > > > > > Regards,
> > > > > >
> > > > > > Authors

---

### Official Review · Reviewer_Chas · 2024-11-09

**Soundness:** 3
**Presentation:** 3
**Contribution:** 3
**Rating:** 5
**Confidence:** 2

**Summary:**

The paper explores Reinforcement Learning from Personalized Human Feedback (RLPHF), introducing a new black-box approach for generating personalized outputs. This approach includes a lightweight Preference Control Model (PCM) that dynamically adjusts next-token prediction weights based on a user’s preference description and the current context. The method requires access to the top output logits of each expert model, bypassing the need for direct weight access. A reward model for each preference dimension, alongside the REBEL online reinforcement learning algorithm, is employed to train the PCM to optimize the average reward for specified preferences. MPD outperforms prior preference-merging techniques in the experiments.

**Strengths:**

The method operates as a black-box, requiring only top logits rather than access to expert model weights.

The method is a scalable solution for personalization by using token-level merging, which reduces the need for extensive fine-tuning.

MPD demonstrates faster performance than previous weight-merging techniques on most modern computing resources.

**Weaknesses:**

Details about the architecture of the Preference Control Model (PCM) are limited and could be expanded.

The MPD training approach appears conventional, lacking novelty in this aspect.

The paper does not assess MPD's performance with larger or alternative LLM architectures beyond Tulu-7B.

The code is not open source, which limits reproducibility. The reasons for its performance advantages over Personalized Soup are not clearly explained.

**Questions:**

In Figure 1, which components represent the expert models? Are these expert models distinct or identical?

Do the expert models utilize different prompts to reflect varied preferences?

If the expert models are identical, what advantages does MPD offer over using a single model for preference alignment?

---

> ### Author Response · Authors · 2024-11-22
> **Response to Reviewer Chas**
>
> Thank you for your constructive feedback and pointing out the scalability of our approach! We address individual points below.
>
> > Details about the architecture of the Preference Control Model (PCM) are limited and could be expanded.
>
> Sorry for the confusion. In Section 4.2, we describe the architecture of PCM, which is a 160M Llama-based model. The small size is attributed to a small hidden size and fewer number of layers compared to Llama 2. The last linear layer of this model is replaced with a much smaller linear layer of size number of preference dimensions ($n = 3$).
>
> > The MPD training approach appears conventional, lacking novelty in this aspect.
>
> We actually intend to make MPD training simple and any online RL algorithm such as PPO and REBEL can be applied for MPD training. Our main contribution is a framework and proof-of-concept where a small PCM that controls how output probability from experts are merged can achieve strong results for multi-objective personalization.
>
> > The code is not open source, which limits reproducibility. The reasons for its performance advantages over Personalized Soup are not clearly explained.
>
> We will make sure to open source our code in the final version of our paper. Both Personalized Soup and MPD use the same set of experts that specialize in individual preference dimensions. The difference is that Personalized Soup averages model **parameters** and MPD **learns** to dynamically mix model **output distribution** at the token level (i.e., mixing weights can be different at different decoding steps). Our results demonstrate that dynamically mixing the model output distribution at token level performs better. Personalized Soup averages the model parameters uniformly regardless of the prompt; in contrast, MPD learns to mix outputs based on preference and prompts. The ability to adapt to different prompts and mix at token level dynamically makes MPD outperforms Personalized Soup.
>
> > In Figure 1, which components represent the expert models? Are these expert models distinct or identical?
>
> Sorry for the confusion; the expert models are denoted as $M_i$ (LLM personalized for X) in Figure 1. These expert models are frozen throughout the training of MPD and can either be identical or distinct as long as they have the same tokenization. In our experiments, these experts have the same architecture.
>
> > Do the expert models utilize different prompts to reflect varied preferences?
>
> Thank you for the question! No, each expert model always uses a fixed preference prompt regardless of varied preferences. The preference prompt for each expert can be found in Table 1.
>
> > If the expert models are identical, what advantages does MPD offer over using a single model for preference alignment?
>
> Thank you for the great question. The expert models have identical architecture but not identical weights. Compared to the single instruction-tuned base model, as seen in Table 2, MPD offers significantly better personalized results than Preference Prompting. Compared to alternative RLHF approaches that directly finetune the base model, MPD is much more lightweight and can be used with blackbox base models since it only finetunes a small PCM module and does not assume weight access of base models.

---

> > ### Comment · Reviewer_Chas · 2024-11-26
> >
> > Thank you for the clarification.
> >
> > Regarding the implementation of PCM, why not simply use BERT or a larger model?
> >
> > Have you tried an inference cost comparison?

---

> > > ### Author Response · Authors · 2024-11-27
> > > **Response to Reviewer Chas**
> > >
> > > Dear Reviewer Chas,
> > >
> > > Thank you for taking the time reading our response and we are happy to answer the questions above.
> > >
> > > > Regarding the implementation of PCM, why not simply use BERT or a larger model?
> > >
> > > Thank you for the suggestion. BERT or other bidirectional encoder models use non-causal attention. Because of this, the output and intermediate layer representation of a token depends on the future tokens yet to be decoded. Whenever a new token is decoded, the representation of past tokens for PCM needs to be recalculated, which causes more computational overhead.
> > >
> > > For a decoder-only model implementation of PCM, the representation of past tokens can be reused. Therefore, we choose a decoder-only Llama-based PCM architecture.
> > >
> > > > Have you tried an inference cost comparison?
> > >
> > > Thank you for the great question. On Line 489-506 of the paper, we conducted an experiment benchmarking the inference speed between Personalized Soup and MPD. Although MPD or in general output merging based approaches require more forward pass compared to weight merging based counterparts such as Personalized Soup, MPD is actually faster than Personalized Soup on average. Intuitively, this is because Personalized Soup cannot batch requests from different preferences whereas MPD can batch on the individual preference dimensions, which has better parallelism. In Appendix A.3, we also provide a coarse theoretical analysis and show that MPD inference speed scales favorably compared to Personalized Soup.
> > >
> > > Please let us know if there is anything we can clarify further.
> > >
> > > Regards,
> > >
> > > Authors

---

> > > > ### Author Response · Authors · 2024-12-01
> > > > **Follow up**
> > > >
> > > > Dear Reviewer Chas,
> > > >
> > > > We thank you for your time and feedback, and would be happy to answer any further questions you may have before the discussion period ends. Please let us know if any issues remain and/or if there are any additional clarifications we can provide.
> > > >
> > > > If you are satisfied with our rebuttal, we would appreciate it if you could reconsider your score.
> > > >
> > > > Regards,
> > > >
> > > > Authors

---

> > > > > ### Comment · Reviewer_Chas · 2024-12-02
> > > > >
> > > > > For the PCM, it would be better to try some different models, like Llama 1B models.

---

> > > > > > ### Author Response · Authors · 2024-12-02
> > > > > > **Response to Reviewer Chas**
> > > > > >
> > > > > > Dear Reviewer Chas,
> > > > > >
> > > > > > Thank you for the great suggestion again. We agree with the reviewer that different models for PCM should be tried. We did not specifically experiment with Llama 3.2 1B because it was just released before the submission deadline and its tokenization is incompatible with Tulu-7B (Llama 2 based tokenization that expert models use).
> > > > > >
> > > > > > In our preliminary experiments, we experimented with PCM being a Tulu-7B model, which is the same size as the expert models. We observed that this increases the training time of our approach while it does not lead to additional improvements on GPT4 win rate. The win rate against Vanilla Prompting, Preference Prompting and Personalized Soup are 81.66%, 53.85% and 52.81% respectively. We hypothesize that this is because PCM only needs to output a vector of dimension 3 and the risk of overfitting increases when PCM is very large.
> > > > > >
> > > > > > We thank the reviewer for the suggestion and will incorporate the above results into the final version of the paper.
> > > > > >
> > > > > > Regards,
> > > > > >
> > > > > > Authors

---

### Author Response · Authors · 2024-11-22
**General Response to Reviewers**

We thank all the reviewers for their time and thoughtful feedback. We appreciate that the reviewers find our problem setting **interesting** and **novel** (XXCM, DVt6), our approach **scalable** and **efficient** (Chas, DVt6, 6WP3), and our paper is **well-written** (XXCM, 6WP3).

We would like to highlight that our experimental setup follows from our related work [1] (Personalized Soup) which is one of the first works on multi-objective personalization for LLMs. We adopt it for a fair and direct comparison.

We respond to each reviewer's valuable critiques in our individual responses. We hope to continue this valuable discussion during the remaining discussion period. Thank you again for your valuable input!

---

### Meta-Review · Area_Chair_HbNE · 2024-12-18

**Metareview:**

The paper proposes a framework for aligning large language models with personalized human preferences. It employs a lightweight Preference Control Model to combine the outputs of expert LLMs at the token level based on user-defined preferences.

The reviewers generally found the problem setting of personalized LLMs interesting and the paper well-written. However, reviewers raised concerns about the necessity and practicality of the proposed method, particularly regarding the availability of black-box expert models and scalability to a diverse range of preferences. Reviewers also noted the lack of technical novelty in key aspects, limited experimental scope (e.g., only Tulu-7B models). The paper requires more work given these concerns raised.

**Additional Comments On Reviewer Discussion:**

The authors provided responses to the reviewers’ questions, clarifying PCM design, reward calculation, and experimental setup. Despite these efforts, the reviewers maintained significant concerns about the paper’s contribution and applicability. Reviewer XXCM and Chas remained critical of the novelty and real-world applicability, while DVt6 and 6WP3 leaned toward marginal acceptance but acknowledged limitations.

---

### Decision · Program_Chairs · 2025-01-22

Reject